# Comparison of Antioxidant Properties of Dehydrolutein with Lutein and Zeaxanthin, and their Effects on Cultured Retinal Pigment Epithelial Cells

**DOI:** 10.3390/antiox10050753

**Published:** 2021-05-10

**Authors:** Małgorzata B. Różanowska, Barbara Czuba-Pelech, John T. Landrum, Bartosz Różanowski

**Affiliations:** 1School of Optometry and Vision Sciences, Cardiff University, Cardiff CF24 4HQ, Wales, UK; 2Cardiff Institute for Tissue Engineering and Repair (CITER), Cardiff University, Cardiff CF24 4HQ, Wales, UK; 3Department of Biophysics, Faculty of Biochemistry, Biophysics and Biotechnology, Jagiellonian University, 30-387 Kraków, Poland; Barbara.Czuba-Pelech@uj.edu.pl; 4Department of Chemistry and Biochemistry, Florida International University, Miami, FL 33199, USA; LandrumJ@fiu.edu; 5Institute of Biology, Pedagogical University, 30-084 Kraków, Poland; Bartosz.Rozanowski@up.krakow.pl

**Keywords:** carotenoid, lutein, zexanthin, dehydrolutein, retina, retinal pigment epithelium, singlet oxygen, photosensitized oxidation, age-related macular degeneration

## Abstract

Dehydrolutein accumulates in substantial concentrations in the retina. The aim of this study was to compare antioxidant properties of dehydrolutein with other retinal carotenoids, lutein, and zeaxanthin, and their effects on ARPE-19 cells. The time-resolved detection of characteristic singlet oxygen phosphorescence was used to compare the singlet oxygen quenching rate constants of dehydrolutein, lutein, and zeaxanthin. The effects of these carotenoids on photosensitized oxidation were tested in liposomes, where photo-oxidation was induced by light in the presence of photosensitizers, and monitored by oximetry. To compare the uptake of dehydrolutein, lutein, and zeaxanthin, ARPE-19 cells were incubated with carotenoids for up to 19 days, and carotenoid contents were determined by spectrophotometry in cell extracts. To investigate the effects of carotenoids on photocytotoxicity, cells were exposed to light in the presence of rose bengal or all-*trans*-retinal. The results demonstrate that the rate constants for singlet oxygen quenching are 0.77 × 10^10^, 0.55 × 10^10^, and 1.23 × 10^10^ M^−1^s^−1^ for dehydrolutein, lutein, and zeaxanthin, respectively. Overall, dehydrolutein is similar to lutein or zeaxanthin in the protection of lipids against photosensitized oxidation. ARPE-19 cells accumulate substantial amounts of both zeaxanthin and lutein, but no detectable amounts of dehydrolutein. Cells pre-incubated with carotenoids are equally susceptible to photosensitized damage as cells without carotenoids. Carotenoids provided to cells together with the extracellular photosensitizers offer partial protection against photodamage. In conclusion, the antioxidant properties of dehydrolutein are similar to lutein and zeaxanthin. The mechanism responsible for its lack of accumulation in ARPE-19 cells deserves further investigation.

## 1. Introduction

3′-Dehydrolutein, ((3R,6′R)-3-hydroxy- β,ε-caroten-3′-one also sometimes referred to as 3′-oxolutein or 3′-ketolutein; Figure 1) has been identified in substantial concentrations in human and monkey liver, blood serum, and ocular tissues [1,2,3,4,5,6,7,8,9,10]. It has been suggested that dehydrolutein is a product of oxidative transformation of dietary carotenoids, lutein (β, ε-carotene-3,3′-diol; Figure 1), and zeaxanthin (β, β -carotene-3,3′-diol; Figure 1). Out of the approximately 40 carotenoids present in a typical human diet, lutein and zeaxanthin are the only carotenoids of dietary origin that accumulate in the retina [11,12,13,14,15]. Antioxidant properties of lutein and zeaxanthin have been widely investigated, and it has been well documented that they are very efficient quenchers of excited electronic states of photosensitizers and singlet oxygen, and can act as scavengers of free radicals [15,16,17].

The ability of lutein, zeaxanthin, and their derivatives to quench electronically excited states is of physiological importance because these carotenoids accumulate in the areas of the human body, such as the skin and eye, where, due to exposure to ultraviolet or visible light in the presence of photosensitisers, generation of excited electronic states of photosensitizers and singlet oxygen can occur [18,19,20,21,22]. Moreover, in the macular part of the retina, where lutein and zeaxanthin accumulate in particularly high concentrations, they can act as optical filters absorbing incoming blue light, thereby preventing blue light from reaching the parts of photoreceptive neurons and retinal pigment epithelium (RPE), where visual pigments are present and potent photosensitizers can accumulate [14,23,24,25,26,27,28,29,30,31]. Therefore, the macular carotenoids are credited with improving visual functions and protecting photosensitive parts of the retina from blue-light-induced oxidative damage. Altogether, the antioxidant and blue-light filtering properties of lutein and zeaxanthin are believed to play an important role in protecting the retina against oxidative stress, and, therefore, in protecting from the development and progression of age-related macular degeneration [15,16,32,33]. Macular degeneration is the primary cause of blindness in the elderly in developed countries.

It has been determined that about 15% to 25% of retinal carotenoids are present in the photoreceptor outer segments, and, in smaller concentrations, in the RPE [34,35]. The photoreceptor outer segments are the parts of photoreceptive neurons where visual pigments are present, and where, following absorption of light, all-*trans*-retinal can be released from photoactivated visual pigments and accumulate in high concentrations [21,34,35,36]. All-*trans*-retinal is a potent photosensitizer with an absorption spectrum that extends into the visible range. In the presence of oxygen, 30% of photons absorbed by all-*trans*-retinal can be used for photosensitized generation of a singlet oxygen [20]. Therefore, the ability of lutein, zeaxanthin, and their derivatives to quench singlet oxygen deserves particular attention. Lutein and zeaxanthin each have the ability to deactivate singlet oxygen with the bimolecular rate constants approaching the diffusion-controlled limits. The deactivation of the singlet oxygen excited states by carotenoids proceeds mainly via a safe route of energy transfer from the excited state of molecular oxygen to the carotenoid molecule, which is followed by thermal relaxation of the carotenoid triplet state to the ground state. This releases the excess energy in the form of heat [37].

Dehydrolutein is present in the human macula at substantial concentrations and increases with age [1,3,4,5,8,10,38]. Post-mortem quantification of dehydrolutein in 4-mm diameter trephine retinal biopsies centred on the maculae from cadavers < 48 years of age (average age of 32+/−8, n = 35) revealed 0.6+/−0.5 ng (0.048 ± 0.040 ng/mm^2^). In people > 48 years of age (average age of 68+/−7, n = 38), it reaches a concentration of 1.3+/−1.8 ng in an identical retinal area (0.10 ± 0.14 ng/mm^2^), with the highest total value detected being about 12.5 ng [1,3,38]. Yet, the antioxidant properties of dehydrolutein and its effects on cultured RPE cells have not been investigated. Therefore, the aim of this study was to compare the abilities of dehydrolutein, lutein, and zeaxanthin to (i) quench singlet oxygen, (ii) protect from photosensitized oxidation of lipids, and (iii) compare their effects on RPE cells in the absence and presence of photosensitizers and light.

## 2. Materials and Methods

### 2.1. General Chemicals and Reagents

Cholesterol, egg yolk phosphatidylcholine (EYPC), all-*trans*-retinal, 93-(4,5-dimethylthiazolyl-2)-2,5-diphenyltetrazolium bromide (MTT), butylated hydroxytoluene (BHT), Chelex 100, and rose bengal (95%) were obtained from Sigma-Aldrich Chemical Co. Dehydrolutein was synthesized as described previously [39]. Lutein (95% pure with 5% of zeaxanthin) and zeaxanthin were a generous gift from DSM Nutritional Products AG (Basel, Switzerland). The spin probe, 4-protio-3-carbamoyl-2,2,5,5-tetraperdeuteromethyl-3-pyrroline-1-yloxy (mHCTPO), was a generous gift from Professor Howard Halpern, University of Chicago, IL, USA. Unless stated otherwise, all procedures involving photosensitizers and/or carotenoids were performed under dim light.

### 2.2. Determination of Singlet Oxygen Quenching by Dehydrolutein, Lutein, and Zeaxanthin

To determine the rate of singlet oxygen quenching, singlet oxygen was generated by photoexcitation of all-*trans*-retinal solution in benzene with a 5-ns laser pulse of the third harmonic of a Q-switched Nd:YAG laser (Continuum Surelite II-10, Photonic Solutions Plc., Edinburgh, UK), and monitored by detecting its infrared emission at 1270 nm by a liquid-nitrogen-cooled germanium diode (Applied Detectors Co., Fresno, CA, USA), connected to an Agilent 54830B digitizer (Agilent Technologies UK Ltd., Cheadle, UK) and analyzed by a RISC workstation operating the LKS.60 nanosecond time-resolved laser photolysis spectrometer (Applied Photophysics, Leatherhead, UK) [19,20,40]. The absorbance of 5.3 µM all-*trans*-retinal in a 1-cm square cuvette was about 0.17 at the excitation wavelength of 355 nm for all the solutions. The rate constant for the quenching of singlet oxygen by carotenoids was determined by quantifying the rate of decay of its infrared phosphorescence in the presence of different carotenoid concentrations [18].

### 2.3. Preparation of Lipid Vesicles (Liposomes)

Multi-lamellar liposomes were prepared either from EYPC or a mixture of EYPC, DMPC, and cholesterol in the presence and absence of all-*trans*-retinal or carotenoids [41,42]. The compounds were dissolved in chloroform in a round-bottom flask, and then the chloroform was evaporated under a stream of argon to form a lipid film. To ensure a complete removal of chloroform, the films were dried under vacuum for at least 1 h. Then the lipid films were hydrated in phosphate buffered saline (PBS). PBS was prepared in deionized glass distilled water and treated with chelating resin, Chelex 100, in order to remove any contaminating metal ions that may catalyse the decomposition of peroxides. For experiments with all-*trans*-retinal, carotenoids were added to the suspension of liposomes from their stock solutions in DMSO, giving the final concentration of 1% DMSO and up to 0.04 mM carotenoids.

### 2.4. Comparison of the Effects of Dehydrolutein, Lutein, and Zeaxanthin on Photosensitized Oxidation of Lipids

To determine the effect of carotenoids on photosensitized oxidation mediated by rose bengal or all-*trans*-retinal, electron spin resonance (ESR) oximetry was used as described previously [41,43,44]. In short, oxygen concentrations were monitored by spectral characteristics of 0.1 mM mHCTPO employed as a nitroxide spin probe. The suspensions of lipid vesicles in the presence of rose bengal or all-*trans*-retinal selected concentrations of carotenoids, and 0.1 mM mHCTPO were irradiated in situ in a flat quartz cell (Wilmad Glass. Co., Buena, NJ, USA) in a resonant cavity of the ESR spectrometer, at an ambient temperature using a 150 W xenon arc lamp (Oriel Corporation, Stratford, CT, USA; 06497 Model 60100) equipped with a combination of lenses and filters (a 5 g/L copper sulphate solution with a 10-cm optical pathlength, a glass cut-off filter, and an interference filter). For experiments with all-*trans*-retinal, a cut-off filter absorbing light < 390 nm, and an interference filter transmitting light of 404 ± 6 nm was used. The irradiance inside the resonant cavity, measured with a photodiode PD Irradiance Meter (Hamamatsu, Photonics, K.K., Hamamatsu City, Japan), was 8.1 mW/cm^2^. For experiments with rose bengal as a photosensitizer, a cut-off filter absorbing light < 520 nm and an interference filter of 542 ± 4 nm were used. The irradiance of the sample was 14.0 mW/cm^2^. The temperature variation was minimized by continuous flow of gaseous nitrogen through the resonant cavity. ESR measurements were performed using the Bruker ESP 300E spectrometer operating at the X band (Bruker, Rheinstetten, Germany). The instrument settings were: microwave power of 1 mW, modulation amplitude of 0.1 G, sweep width of 3.0 G, time constant of 10.24 ms, and conversion time of 20.48 ms.

### 2.5. Cell Culture and Feeding with Carotenoids

ARPE-19 cells, which are a spontaneously “immortalized” and well-characterized human retinal pigment epithelial cell line derived from a 19-year-old male donor, were purchased from the American Type Culture Collection (ATCC, Manassas, VA, USA) [45]. ARPE-19 cells were routinely passaged by dissociation in 0.05% (*w/v*) trypsin, maintained at 37 °C in a humidified incubator filled with 5% CO_2_ in air, and fed every two to three days with minimal essential medium (MEM, Sigma-Aldrich Chemical Co., St Louis, MO, USA), containing 10% heat-inactivated foetal calf serum (FCS), L-glutamine, and penicillin-streptomycin (Sigma-Aldrich Chemical Co., St Louis, MO, USA). All experiments were performed on confluent cell monolayers, passage numbers between 25 and 29, seeded either in 75 or 25 cm^2^ flasks, or in 24-well or 12-well plates [19,46,47].

### 2.6. Administration of Carotenoids to Cultures of RPE Cells

Stock solutions of carotenoids were prepared under argon in DMSO under dim light. To facilitate binding of carotenoids to serum lipoproteins and albumin, the carotenoid solution was added to FCS and incubated for 1 h at 37 °C with mixing. Then, the carotenoid-enriched FCS was added to the culture medium, giving a final concentration of 2 μM carotenoids, 0.2% DMSO, and 10% FCS. Confluent monolayers of cells, starting usually 10 days after seeding, were fed with freshly prepared carotenoid-enriched culture medium three times a week for up to 19 days.

Alternatively, carotenoids were injected directly into Dulbecco’s PBS with calcium and magnesium (DPBS), and incubated with cultured cells for 60 min during exposure to photosensitizers and light. After the incubation, cells were washed with DPBS, and then DPBS was replaced with the culture medium for a further culture or the reductive activity assay.

### 2.7. Evaluation of Carotenoid Content in Cells

To determine carotenoid uptake by ARPE-19 cells, the cells were trypsinized, 40 µL aliquots were taken for cell counting, and the remaining cell suspensions were mixed at a 5:8 ratio with chloroform/methanol (2:1, *v/v*) containing 1 mM BHT to prevent oxidation [47]. The mixture was vortexed, and followed by centrifugation to separate the phases. The bottom chloroform-rich phase was removed, replaced by a chloroform phase obtained by centrifugation of a mixture of PBS with chloroform and methanol, and the extraction was repeated. Combined chloroform phases were dried under nitrogen and resolubilized in acetone or chloroform prior to spectrophotometric detection of the accumulated carotenoids. The absorption spectra of solubilized extracts were measured in a 0.8-mL cuvette with a 1-cm optical pathlength.

### 2.8. Evaluation of Cell Viability

The morphology of the RPE monolayer was routinely observed using the inverted microscope before the media changes and after experimental treatments. Cell viability was quantified by measurements of their reductive activity by the MTT assay, and after trypsinization by cell counts [19,46,47]. Cells were counted in a Bürker chamber. For the MTT assay, cells were washed with DPBS, and then incubated for 60 min in serum-free MEM containing 0.5 mg/mL MTT. Then, the cells were washed with DPBS and solubilized with isopropanol acidified with 0.5% HCl. Optical density of the solubilised formazan was read at 570 nm. The reductive activities of cells exposed to carotenoids in the absence of photosensitizers are expressed relative to DMSO-treated cells from the same plate and given as a percentage. The reductive activities of cells exposed to DMSO/carotenoids in the presence of photosensitizers are expressed as ratios to cells exposed to DMSO/carotenoids in the absence of photosensitizers on the same plate, and given as a percentage.

### 2.9. Exposure of RPE Cells to Light and Photosensitizers

To test whether carotenoids can affect phototoxicity, ARPE-19 cells were washed from the culture medium with DPBS, and the medium was replaced with DPBS containing 0.5 μM rose bengal or liposomes containing 0.5 mM all-*trans*-retinal and 2.8 mg/mL EYPC. The 24-well plates with cells were placed on top of a glass sheet above the set of fluorescent tubes, and cells were irradiated at 24–26 °C with white fluorescent light for 20, 30, 40, or 60 min. The irradiance, measured with IL2000 Spectrotube spectroradiometer (International Light Inc., Newbury, MA, USA), was 0.46 mW/cm^2^. After selected irradiation times, the cells were washed with PBS and fed with culture medium. The MTT assay was performed immediately after exposure, or 24 h after exposure.

### 2.10. Subcellular Fractionation and Transmission Electron Microscopy (TEM)

The subcellular fractions from cells supplemented for 19 days with lutein or zeaxanthin were isolated as described previously [20,44,48,49,50,51,52]. In short, confluent ARPE-19 monolayers were scraped from flasks (each with a surface area of 75 cm^2^). Cells from six flasks for each carotenoid were combined and homogenized, which was followed by centrifugation at 60 g for 7 min in PBS with 1 mM EDTA to remove nuclei and cell debris. The pellet was subjected to another homogenization, while the supernatant was centrifuged at 6000 g for 10 min. The pellets from 6000 g centrifugations were combined, resuspended in 0.3 M sucrose, and layered on top of a discontinuous sucrose gradient consisting of 2.00, 1.80, 1.60, 1.55, 1.50, 1.40, 1.20, and 1.00 M sucrose. The samples were ultra-centrifuged using L7 Beckman ultra-centrifuge for 1 h at 103,000 g. The fractions collected from each sucrose density interface were diluted in 0.1M phosphate buffer at a pH of 7.2 and spun down for 10 min at 6000 g. The resuspended pellets were divided into two equal portions: one for carotenoid extraction and the other for TEM. All procedures were done on ice under dim light, and centrifugations were done at 4 °C.

The processing of subcellular fractions for TEM was performed as described previously [53]. The pellets were resuspended in 2.5% glutaraldehyde in 0.1M phosphate buffer at a pH of 7.2 at 4 °C, washed, solidified in 2% agar solution at 4 °C, postfixed in 1% OsO_4_, dehydrated in graded series of ethanol and propylene oxide solutions, and embedded in Epon 812. Ultrathin sections were cut on Tesla BS 490A ultramicrotome, double stained in uranyl acetate and lead citrate, and examined with a Tesla BS-500 electron microscope.

### 2.11. Statistical Analysis

Unless stated otherwise, results are expressed as means ± SDs from at least three independent experiments. Statistical analyses were performed using SigmaPlot14 and one-way ANOVA, and were followed by all pairwise multiple comparison procedures using Hol-Sidak method, where *p* < 0.05 was considered statistically significant.

## 3. Results

### 3.1. Singlet Oxygen Quenching

To compare the ability of dehydrolutein to that of lutein or zeaxanthin to quench singlet oxygen, we used a 5-ns laser pulse excitation of all-*trans*-retinal to generate singlet oxygen and monitored the rate of its decay in the absence and presence of increasing concentrations of dehydrolutein, lutein, or zeaxanthin (Figure 2A). For all three carotenoids, the rates of singlet oxygen decay increased linearly with increasing carotenoid concentration (Figure 2B). This allowed us to calculate the singlet oxygen quenching rate constants for each carotenoid. Our results demonstrated that lutein and zeaxanthin exhibited quenching rate constants of (0.55 ± 0.02) × 10^10^ and (1.23 ± 0.02) × 10^10^ M^−1^s^−1^, respectively, which are similar to the values reported in literature. Previously reported values for singlet oxygen quenching in benzene are 1.2 × 10^10^ M^−1^s^−1^ and 0.66 × 10^10^ M^−1^s^−1^ for zeaxanthin and lutein, respectively [37]. The rate constant of singlet oxygen quenching by dehydrolutein, (0.77 ± 0.02) × 10^10^ M^−1^s^−1^ was intermediate between the values for lutein and zeaxanthin, demonstrating that dehydrolutein is also an effective singlet oxygen quencher.

### 3.2. Effects of Dehydrolutein, Lutein, and Zeaxanthin on Photosensitized Oxidation

We have shown previously that zeaxanthin can inhibit lipid oxidation when incorporated into lipid vesicles (liposomes) and exposed to rose bengal and green light [41]. Here, we used a similar model system to compare effects of dehydrolutein, lutein, and zeaxanthin on rose-bengal-mediated oxidation (Figure 3). Liposomes containing unsaturated lipids in the absence or presence of carotenoids were exposed to rose bengal and light to induce photo-oxidation, which was monitored by measuring oxygen consumption. At 10 µM concentrations, all three carotenoids significantly decreased the rate of oxygen consumption. Zeaxanthin showed the greatest effectiveness in slowing down the initial rate of oxidation by 47%, which was followed closely by dehydrolutein and which decreased the rate by 45%. However, that 2% difference was not statistically significant. Lutein at a concentration of 10 µM was the least effective of the three carotenoids producing a decrease in the rate of oxidation by 32%, and its effect was statistically different from that of both dehydrolutein and zeaxanthin (*p* = 0.005 and 0.002, respectively). Increasing concentrations of carotenoids to 20 µM brought about greater decreases in the rates of oxidation by 60%, 67%, and 69% for lutein, dehydrolutein, and zeaxanthin, respectively, with no statistically significant differences between any of the carotenoids. At 40 µM concentrations, the carotenoids decreased the initial rate of oxidation by 97% to 98%, and all three carotenoids were similarly effective.

Rose Bengal is a water-soluble photosensitizer, which associates with the lipid membrane, but does not penetrate through it [46]. Its photochemical properties are well-characterized and its effects on cultured ARPE-19 cells have been previously tested [46]. It is used in ophthalmic practice for diagnostic purposes on the surface of the eye and as a potential agent for corneal photo-crosslinking. It does not penetrate inside the eye and has no physiological effects on the retina [54]. To compare the effect of the three carotenoids on photo-oxidation under more physiologically relevant conditions, we used all-*trans*-retinal, which is a form of vitamin A that can reach high levels in the photosensitive structures of the retina (Figure 4). As mentioned already, all-*trans*-retinal is a lipophilic photosensitizer that absorbs ultraviolet, violet, and blue light and can accumulate in the retina in photoreceptor outer segments during the normal physiological mechanism of retinal light detection [21,36,49,55,56]. Under the experimental conditions used, in the absence of carotenoids, all-*trans*-retinal and blue light induced a 6.5-fold faster oxygen consumption than that induced by rose bengal and green light (Figure 2 and Figure 3). All three carotenoids exhibited effective photoprotection against photoxidation induced by all-*trans*-retinal and blue light at 10 µM concentration (*p* < 0.001, in all three cases), decreasing the rate of oxygen consumption by 30%, 34%, and 45%, respectively, for lutein, dehydrolutein, or zeaxanthin. There was no significant difference between samples containing 10 µM lutein and dehydrolutein. Zeaxanthin was more effective than either lutein or dehydroluten (*p* < 0.001, in both cases) (Figure 4). At 20 µM concentration, lutein decreased the rate of oxygen consumption by 68% and was less effective than dehydrolutein or zeaxanthin (*p* < 0.001, in both cases), which decreased the rates by 72% and 75%, respectively. There was no significant difference in protection offered by either dehydrolutein or zeaxanthin at this concentration. At 40 µM concentration, the carotenoids decreased the rates of oxygen consumption from 88% to 92% and there were no significant differences between any of the carotenoids.

Altogether, the differences between these carotenoid abilities to slow down photooxidation could be seen only at a concentration of 10 µM. Dehydrolutein was more effective than lutein, and similar to zeaxanthin, in protection of lipids from photosensitized oxidation mediated by rose bengal. Protection against all-*trans*-retinal-induced photooxidation by dehydrolutein or lutein was similar, while zeaxanthin was slightly more effective. At 20 and 40 µM concentrations, all three carotenoids were equally effective in protection from photooxidation.

### 3.3. Effects of Supplementation with Dehydrolutein, Lutein, and Zeaxanthin on Carotenoid Content in Cultured Retinal Pigment Epithelial Cells ARPE-19

ARPE-19 cells are a spontaneously immortalized cell line derived from retinal pigment epithelium of the 19-year-old male donor, and were used in passage numbers where these cells retain their retinal pigment epithelial cell characteristics [45]. Importantly, these cells express receptors, such as a class B scavenger receptor type 1 (SR-BI) and low-density lipoprotein (LDL) receptor, which have been demonstrated to be involved in carotenoid uptake [57,58,59,60]. To compare the effects of dehydrolutein with those of lutein and zeaxanthin in cultured ARPE-19 cells, the confluent cells were incubated for up to 19 days with 2 µM carotenoids solubilized first in foetal calf serum (FCS) and then in the culture medium to mimic physiological conditions where carotenoids are carried in the blood plasma mostly bound to lipoproteins. The concentration of carotenoids was chosen based on the highest values of lutein and zeaxanthin in human serum reported in the literature. At the time when lutein and zeaxanthin supplements were not available, it was determined, in a large cohort of 8229 persons above the age of 40 years of various ethnic backgrounds, that the average combined concentration of lutein and zeaxanthin in human serum in people with the highest values was 0.79 µM [61]. It was shown later that, by supplementation, these values can be increased to about 2 µM [12,62,63].

At selected days, cells were trypsinized, 40-µL aliquots were taken for determining cell concentration, and the remaining suspension was used for extracting carotenoids and quantifying by spectrophotometry (Figure 5). Feeding cells with zeaxanthin resulted in a monotonic, almost linear, increase in concentration of cellular zeaxanthin, reaching concentrations of about 1.1 nmol/million cells after eight feedings provided over 19 days (Figure 5). Lutein uptake was less efficient and appeared to reach a plateau after six feedings, and a cellular content of about 0.7 nmol/million cells after 19 days. Based on these data, the average concentrations in cells can be estimated. Assuming the height of the cell monolayer is 8 µm and the surface area of the cell culture flask is 25 cm^2^, the 4.2 million cells occupied a volume of 0.02 mL, giving the average concentration of 0.23 and 0.15 mM for zeaxanthin and lutein, respectively. This means that cells accumulated 116-fold and 74-fold greater concentrations of carotenoids than those provided in the culture medium. This is consistent with previous studies, albeit with a much shorter supplementation time, showing that ARPE-19 cells accumulate lutein and zeaxanthin and can achieve intracellular concentrations exceeding the concentration of these carotenoids in the culture medium by several-fold [59,64]. In contrast, cells fed with dehydrolutein accumulated very little, if any, of this carotenoid (Figure 5). It has been suggested that any dehydrolutein formed in the retina by oxidation of lutein may be reduced to regenerate lutein or epi-lutein [3]. A reduction of keto carotenoid canthanxanthin to its corresponding mono derivatives and diol derivatives has been documented in the primate retina [65]. However, we did not observe accumulation of any carotenoids in cells fed with dehydrolutein. These results indicate a highly selective uptake and accumulation of lutein and zeaxanthin, and a surprising ability of RPE cells to either exclude or efficiently excrete absorbed dehydrolutein.

### 3.4. Effects of Supplementation with Dehydrolutein, Lutein, and Zeaxanthin on Viability of ARPE-19 Cells

Importantly, the supplementation with carotenoids did not affect cell numbers in the confluent cultures. The cell numbers remained similar over the entire period of supplementation in the flasks fed with different types of carotenoids, similar to cell numbers in the flasks treated with DMSO only, and similar to cell numbers per flask prior the supplementation treatment (Figure 6A). The reductive activities of carotenoid-containing cells were significantly reduced in comparison with cells fed with dehydrolutein or DMSO (Figure 6B). The reductive activity of cells fed with lutein or zeaxanthin, measured by the MTT assay, were 18% and 19% smaller, respectively, than reductive activities of cells supplemented with DMSO only (*p* < 0.001 in both cases). Cells fed dehydrolutein had a decreased reductive activity that was 8% lower than cells fed with DMSO, but that difference was not statistically significant (*p* = 0.075).

The 1 h of exposure for ARPE-19 cells to Dulbecco’s phosphate buffered saline (PBS), in the presence or absence of visible light, upregulated the reductive activities of cells fed lutein or zeaxanthin. As a result, the values were similar for all three carotenoids (Figure 6C,D), suggesting that deprivation of glucose and other nutrients present in the normal growth media can upregulate the reductive activity once they are available again. Twenty-four hours post-exposure to Dulbecco’s PBS with and without light, the reductive activities of lutein-containing or zeaxanthin-containing cells returned to the values similar to those observed without any pre-treatment in PBS (Figure 6B,E,F). It has been shown previously on canine lens epithelial cells that lutein can decrease their reductive activities measured by the MTT assay [66]. The reductive activity assay measured by MTT reflects mainly the activity of glycolytic enzymes and NAD(P)H production [67]. It has been shown that lutein can decrease activities of several reductive enzymes, such as aldose reductase, sorbitol dehydrogenase, and isocitrate dehydrogenase [68,69], and that lutein and zeaxanthin can decrease the rate of oxidative phosphorylation [70,71,72,73,74]. However, there are also reports demonstrating that lutein can increase glycolysis and upregulate oxidative phosphorylation [75]. Since the presence of carotenoids affected the reductive activities of cells, the reductive activities of cells exposed to photosensitizers and light were calculated as a ratio of reductive activity of cells exposed to light in the presence of the photosensitizer to reductive activity of cells in the absence of the photosensitizer, but subjected to the same treatment with respect to carotenoids and light.

### 3.5. Effects of Supplementation with Dehydrolutein, Lutein, and Zeaxanthin on Susceptibility of ARPE-19 Cells to Photosensitized Damage

Similar to a previous report [46], exposure to green light in the presence of rose bengal induced a rapid damage to ARPE-19 cells, resulting in an immediate decrease in their reductive activity (and viability) when measured immediately after 1 h of exposure (Figure 7A). The reductive activity in cells treated with DMSO without carotenoids and exposed for 20, 40, and 60 min to rose bengal and light decreased by 21%, 61%, and 76%, respectively, in comparison to cells exposed to light in the absence of rose bengal. The reductive activities of cells fed for 19 days with lutein, zeaxanthin, or dehydrolutein and then exposed to rose bengal and light were similar to those in cells without carotenoids (Figure 6A).

Unlike phototoxicity of rose bengal, which causes immediate loss of cell viability [46], all-*trans*-retinal phototoxicity takes longer to affect cell viability [19], and, therefore, the reductive activities were measured 24 h after the exposure. The reductive activities of cells exposed to all-*trans*-retinal and light were decreased in comparison to cells without all-trans-retinal by about 32% and 78% for 30-min and 60-min exposures, respectively (Figure 7B). Responses of cells that were supplemented with lutein, zeaxanthin, dehydrolutein, or DMSO only were not significantly different from each other. The carotenoids accumulated over 19 days of feeding did not make cells more resistant to photosensitized killing by either rose bengal or all-*trans*-retinal when compared to cells without carotenoids (Figure 7A,B). It appears that incorporation of lutein and zeaxanthin into cells makes them ineffective in protecting the cells from photosensitizers, which generate their reactive oxygen external to the plasma membranes.

This may suggest that the massive accumulation of lutein and zeaxanthin occurs in intracellular organelles. Following endocytosis of lutein and zeaxanthin, it can be expected that these carotenoids enter an endo-lysosomal compartment. To determine whether or not lutein and zeaxanthin accumulate in ARPE-19 cell endosomes/lysosomes, the cells were cultured in 75-cm^2^ flasks and supplemented with lutein and zeaxanthin for 19 days. For each carotenoid, cells were cultured in six flasks to give a similar surface area of the RPE monolayer as RPE-occupying hemispheres of 40 human eyes we used for subcellular fractionation in our previous studies [44,48,49,50,51,52]. To investigate the localization in the ARPE-19 cell, we performed subcellular fractionation using differential centrifugation. Most lutein and zeaxanthin were lost in the pellet from low-speed centrifugation and/or in the supernatant from pelleting the post-nuclear fraction. The post-nuclear fraction was separated by ultracentrifugation on a discontinuous sucrose gradient. Lutein and zeaxanthin were detectable at interfaces at several sucrose densities starting at 0.25/1.0 M down to 1.4 M (Figure 8A). Examination of these bands by transmission electron microscopy (TEM) revealed that these fractions were rich mostly in endosomes, lysosomes, phagolysosomes, and some mitochondria (Figure 8B–E).

To determine whether carotenoids can protect the cells against photodamage while localised extracellularly, cells pre-treated with carotenoids for 19 days were exposed to rose bengal and light in the presence of 2-µM carotenoids and 0.2% DMSO solubilized directly in Dulbecco’s PBS (Figure 9A). This treatment resulted in a significant protection by carotenoids at all exposure times. In case of a 20-min exposure to light and rose bengal, the reductive activity for DMSO-treated cells was 77%, whereas, for cells treated with dehydrolutein, lutein, and zeaxanthin, these activities were significantly higher: 95%, 92%, and 90%, respectively. The 40-min exposure to light and rose bengal resulted in a reductive activity of 43% in DMSO-treated cells, and 64%, 67%, and 66% for dehydrolutein, lutein, and zeaxanthin-treated cells. In addition, for the 60-min exposure, the reductive activities for carotenoid-treated cells were statistically greater than for DMSO-treated cells: 25% in DMSO-treated cells, and 40%, 41%, and 38% for cells supplemented with dehydrolutein, lutein, or zeaxanthin, respectively. This approach also proved effective in protecting against photodamage induced by all-*trans*-retinal and light, where carotenoids increased reductive activities from 25% in DMSO-only-treated cells to about 36%, 34%, and 33%, respectively, for cells supplemented with lutein, zeaxanthin, or dehydrolutein and exposed to the retinoid and light for 60 min (Figure 9B). For cells exposed to all-*trans*-retinal and light for 30 min, the increase was from 67% in DMSO-treated cells to 87%, 88%, and 86% for cells treated with dehydrolutein, lutein, or zeaxanthin, respectively. While the *p* values were smaller than 0.001 for differences between DMSO and a carotenoid treatment for each exposure type, there were no statistically significant differences between any of the three carotenoids.

There are numerous reports showing that lutein and/or zeaxanthin can modulate expression of several antioxidant enzymes, and can activate the nuclear factor erythroid 2-related factor 2 (Nrf2) signalling responsible for the expression of antioxidant response element (ARE) genes, and, subsequently, the synthesis of antioxidant and detoxification proteins [68,69,76,77,78,79,80,81,82,83,84,85,86,87,88,89]. It has been reported that DMSO can activate Nrf2 [90]. Therefore, we considered a possibility that long-term feeding with carotenoids/DMSO may affect cellular responses to oxidative stress induced by the exposure to light and photosensitizers, and used ARPE-19 cells without the long-term feeding with carotenoids while exposing them to photosensitizers and light with DMSO/carotenoids provided at the same time (Figure 9C,D). Under these conditions, the reductive activities were similar as in cells supplemented with carotenoids for 19 days and then exposed to photosensitizers and carotenoids solubilized in PBS (compare Figure 9A with Figure 9C, and Figure 9B with Figure 9D). For the 60-min exposures, the reductive activities increased from about 22% in cells treated with DMSO to 42%, 43%, or 42% for cells exposed to light and rose bengal, and from 20% to 36%, 35%, and 32% for cells exposed to light and all-*trans*-retinal in the presence of dehydrolutein, lutein, or zeaxanthin, respectively (Figure 9C,D). Again, there were no significant differences in protection offered by the three carotenoids under the same exposure conditions in any of these cases.

The protection offered by the short-term treatment with carotenoids was similar in cells with and without long-term carotenoid supplementation (Figure 9A–D). Therefore, it appears that the accumulation of carotenoids inside cells does not affect cellular responses to the photo-oxidative damage induced by extracellular photosensitizers and light.

## 4. Discussion

Altogether, our results demonstrate that antioxidant properties of dehydrolutein are similar to those of lutein and zeaxanthin. All three carotenoids are effective singlet oxygen quenchers, which can slow down the photo-oxidation of lipids mediated by photo-excited rose bengal or all-*trans*-retinal. This is important because at least some zeaxanthin and lutein in the retina appear to be bound with high selectivity and specificity to glutathione S-transferase Pi isoform (GSTP1) and steroidogenic acute regulatory domain protein 3 (StARD3) proteins, respectively [8,91,92,93,94,95], which would limit their diffusion rates [96] and, consequently, their efficiency as singlet oxygen quenchers. On the other hand, to date, no dehydrolutein-binding protein has been identified. Lacking a specific binding protein, dehydrolutein may be more likely to be present in the lipid membranes as a free carotenoid, and, thereby, may be more physiologically effective as a singlet oxygen quencher protecting components of the lipid membrane from photosensitised oxidation. Therefore, the localization of dehydrolutein in the retina deserves further investigation to determine: (i) in which retinal layers, (ii) in which cell types and their subcellular compartments it is present, and (iii) whether it is present as a free carotenoid in lipid membranes or it is bound to proteins.

These are challenging tasks because it is still unknown in which subcellular compartment lutein and zeaxanthin accumulate, and whether they accumulate in the same compartment or in different compartments. What is well documented is that lutein and zeaxanthin accumulate in the retina in different areas as a function of eccentricity from the fovea centre, with zeaxanthin being highly concentrated in the fovea while lutein being distributed more diffusely across the retina [24,97,98]. It has been determined using confocal resonance Raman microscopy that the zeaxanthin:lutein ratio can be greater than 9:1 in the centre of the fovea, decreases to 4:1 200 µm from the centre, and an additional 200 µm away, the ratio becomes 1:4 [24]. In the peripheral retina, >5 mm from the foveal center, the total concentration of zeaxanthin+lutein is approximately 1/100 that of the central fovea [99]. Across the retinal layers in the fovea, the greatest concentrations of each, zeaxanthin and lutein, are in the Henle’s fibre layer and are clearly visible in microsections [100]. The Henle’s fibre layer contains axons of foveal photoreceptors and these carotenoids are also present in high concentrations where the Henle’s fibre layer transitions into the outer plexiform layer and where axonal terminals of photoreceptors make synaptic connections with bipolar and horizontal cells. Lutein and zeaxanthin also reach high density in the inner nuclear layer with cell bodies of glial Müller cells and three types of neurons: bipolar, horizontal, and amacrine cells in the inner plexiform layer with synaptic connections between bipolar cells with amacrine and a retinal ganglion cell, and in the outer nuclear layer with cell bodies of photoreceptors. The Müller cell processes are spread across all these layers and there is a growing body of evidence that macular carotenoids can be bound to these glial cells (reviewed by Curcio [101]).

It has been demonstrated that the majority of lutein and zeaxanthin in the macula is highly organized and concentrically aligned relative to the central axis, passing through the pupil and the macula, i.e., the visual axis [102]. Such an organizational feature of the macular pigment is the accepted explanation of the entoptical phenomenon, which is known as Haidinger’s brushes. Haidinger’s brushes can be seen as an hour-glass shaped darkening when a horizontally plane-polarized short wavelength (blue) light passes through the retina at the center of the visual field and is attributed to the preference of the concentrically arrayed pigment molecules laying above and below the axis to absorb the horizontal polarized light whereas molecules laying either to the left or right of the axis do not. This hour-glass entopical figure rotates with the plane of the polarized light, always being oriented at 90° to the plane of polarization, and has an action spectrum identical to the absorption spectra of the carotenoids, lutein and zeaxanthin, which compose the macular pigment [102]. The arrangement of lutein and zeaxanthin arises because Henle’s fibres extend radially from the center of the fovea outward to the peripheral retina and the carotenoids are either incorporated by spanning the bilayer lipid membrane or are associated with proteins that are axially arrayed along the axon length, which results in a similar organizational arrangement of the carotenoids.

It has been shown that carotenoids can bind to several proteins [29]. They include tubulin, which is abundant in axons, but the binding is of low specificity and affinity. In addition, the interphotoreceptor retinoid binding protein (IRBP), which is present in the space around inner and outer segments of photoreceptors, binds both lutein and zeaxanthin with low specificity, but the affinity is high, which is similar to that of retinoids [103]. These are constituents of all species and are not the primary candidates responsible for the binding of carotenoids of the macular pigment, a feature that is anatomically unique to the primate retina. In addition to those mentioned above, there are other proteins in the primate retina that bind to carotenoids with high specificity and affinity: steroidogenic acute regulatory domain 3 (StARD3) and a pi isoform of glutathione S-transferase (GSTP1), which bind to lutein and zeaxanthin, respectively [29].

GSTP1 is considered to be a cytosolic enzyme whereas the majority of the lutein and zeaxanthin in the retina are membrane-associated and require detergent to solubilize them [34,92]. However, it cannot be excluded that GSTP1 undergoes post-translational modification, which anchors it in the lipid membrane. It has been reported that the immunofluorescence of the antibody to human recombinant GSTP1 appears throughout all retinal layers in the human macula, including inner and outer segments of photoreceptors, with the greatest expression in the inner and outer plexiform layers, where retinal neurons make synaptic junctions [92]. It appears absent in the nuclei of retinal cells. In another report, the strongest labelling in the monkey macula appears along the outer limiting membrane with some cone inner segments stained heavily while others are not stained at all [29]. It has been proposed that GSTP1 can bind zeaxanthin at the carotenoid to protein stoichiometry of 2:1 [92]. It has been pointed out that GSTP1 is a small, 24 kDa globular protein. Therefore, binding a long planar ligand, such as zeaxanthin, could expose a large portion of that lipophilic ligand to the external environment [104].

StARD3 was detected by Western blotting in the human neural retina and RPE-choroid, and its mRNA expression was confirmed by RT-PCR in the human retina [95]. Immunocytochemistry on the monkey retina demonstrated StARD3 antibody binding in retinal layers spanning from the ganglion cell layer to the outer limiting membrane with the greatest density in the Henle’s fibre layer [29,95]. In addition, RPE was heavily stained but that staining was non-specific to the StARD3 antibody. The retinal layers of the outer segments of photoreceptors containing membraneous discs packed with visual pigments, and inner segments of photoreceptors with mitochondria, endoplasmic reticulum, and Golgi apparatus, as well as the interphotoreceptor matrix filling the space between photoreceptors exhibited relatively low staining intensity [95]. Co-labelling of cone photoreceptors showed that the StARD3 antibody labels cones with greater intensity than rods. However, the labelling on cones was rather patchy with only some areas showing co-labelling. The co-labelling of Müller cells showed no co-localization of the StARD3 antibody with Müller cells.

It has been demonstrated that optical density of the macular pigments can be increased by supplementation with lutein over a period of 140 days [105]. Once the supplementation was stopped, the increase in macular pigment density continued for another 43 days and reached a stable level, which remained the same for the following 150–200 days. Afterward, monitoring was stopped. The long-term stability of the macular pigment in the retina over this period after discontinuation of supplementation is remarkable, considering the environment in the retina, which is exposed daily to high fluxes of light in the presence of various photosensitizers and high abundance of polyunsaturated lipids. Carotenoids are rather labile compounds susceptible to oxidative degradation. Therefore, their stability in the macula also suggest that they are protected from oxidation and degradation by proteins and/or sequestered in other immobilizing structures.

It remains an open question which subcellular compartments accumulate lutein and zeaxanthin, and whether or not they accumulate in the same structures, i.e., cytoplasm, vesicles, membranes, or subcellular protein framework even though, as noted earlier, evidence supports cell lipid bilayers or other radially arrayed structures within the retina. The massive accumulation of lutein and zeaxanthin in cultured ARPE-19 cells prompted us to investigate their localization in these cells using differential centrifugation to quantify carotenoids in subcellular fractions. Lutein and zeaxanthin were detected at interfaces at several sucrose densities starting at 0.25/1.0 M down to 1.4 M. Examination of these bands by transmission electron microscopy revealed that these fractions were rich mostly in endosomes, lysosomes, phagolysosomes, and some infrequent mitochondria. This suggests that lutein and zeaxanthin may co-localize in phagocytosed tips of photoreceptor outer segments and offer their protection from singlet oxygen generated due to photo-excitation of all-*trans*-retinal, oxidized polyunsaturated lipids and/or lipofuscin [19,20,21,44,106,107].

Our results also demonstrate that, while the long-term exposure of ARPE-19 cells to dehydrolutein, lutein, or zeaxanthin leads to a substantial accumulation of lutein and zeaxanthin within the cells, the dehydrolutein does not accumulate to a significant extent. This interesting observation suggests that ARPE-19 cells possess a selective uptake mechanism for lutein and zeaxanthin but can exclude dehydrolutein. Alternatively, there is an efficient mechanism of dehydrolutein removal following uptake. It has been shown that ARPE-19 cells can take up lutein and zeaxanthin from human serum, and the uptake is more efficient for these carotenoids when zeaxanthin is bound to HDLs than to LDLs, and when lutein is bound to LDLs than to HDLs [59]. In the human retina, similar to the results shown here, the selective accumulation occurs. Of approximately 14 dietary carotenoids present in blood serum [91], only two accumulate in the retina [11,12,13,14,15]. The RPE, together with endothelial cells of retinal capillaries, provides the blood-retina barrier, and, therefore, it is involved in the selective uptake of carotenoids and their transepithelial transport [58]. However, the mechanisms responsible for that selectivity and transepithelial transport into the retina are still poorly understood. Further investigations of dehydrolutein interactions with ARPE-19 cells may be helpful as a model to investigate these transport mechanisms and determine if the presence of dehydrolutein in the retina is a result of its uptake from the blood via transepithelial transport or a result of converting of lutein/zeaxanthin inside the retina. It has been shown by Thomas and colleagues that β-carotene-9′,10′-dioxygenase (BCO2), which catabolizes various carotenoids and apo-carotenoids, can convert dehydrolutein into apocarotenoids [108]. It has been reported that ARPE-19 cells express this enzyme using qRT-PCR and Western blotting. Exposure to lutein further increases BCO2 mRNA [109]. The experimental difficulty is that the commercial antibody used to detect BCO2 in the human and mouse retinas by Li and colleagues [110] was shown later to be non-specific to BCO2 and labelled something else in both *bcdo2+/+* and *bcdo2-/-* mice [108]. There is a possibility that could be tested on ARPE-19 cells that lutein and zeaxanthin can avoid enzymatic degradation by binding to proteins whereas dehydrolutein remains unbound and is an available substrate for BCO2. Another pathway by which ARPE-19 cells could remove dehydrolutein is by incorporating them into lipoproteins that they synthesize and secrete. Such a scenario is supported by experiments showing that carotenoid-containing lipoproteins can be synthesized and secreted by CaCo-2 cells [57,111]. Curcio and colleagues demonstrated RPE cells can synthesize and secrete lipoproteins [101,112], while Amin et al. [113] demonstrated that ARPE-19 cells can secrete membranous sub-RPE deposits when supplemented with a retinal extract.

We have shown that cells with accumulated lutein and zeaxanthin exhibit a small decrease in the reductive activity toward MTT. The decrease is reversible. After depriving cells from nutrients present in the culture medium for an hour, the reductive activity increases to levels similar to cells without accumulated carotenoids once the glucose and other nutrients became available. The MTT assay is widely used as a measure of cell viability. However, it measures the reduction of MTT to formazan, which can be affected by an activity of various enzymes and availability of NAD(P)H [67]. It was demonstrated that the MTT reductive activity is not affected, or, in some cases, even enhanced (by sparing NADH), by inhibitors of a mitochondrial electron transport chain, but it decreases in the presence of inhibitors of glucose uptake and cytoplasmic glycolysis [114]. Therefore, it has been suggested that MTT reduction can be viewed as a measure of the glycolytic activity and NAD(P)H production [114]. It has been shown that lutein supplementation can affect activities of several enzymes in the heart and kidney of a rat model with diabetes [68], can modulate expression of a number of genes in murine liver [69], and, in ARPE-19 cells, activates the transcription factor Nrf2, which, in turn, activates genes encoding antioxidative and phase II enzymes that are involved in the maintenance of the cellular redox status as well as in the detoxification of xenobiotics, including NAD(P)H:quinone oxidase, NQO1 [85]. Thus, the decreased reductive activity in the presence of lutein and zeaxanthin may reflect their effects on NAD(P)H and glycolysis. In addition to the MTT assay, we used cell counting as a measure of cell viability. The cell numbers remained similar to the initial ones throughout the duration of the 19-day supplementation. Therefore, we concluded that lutein, zeaxanthin, and dehydrolutein do not affect the cell viability.

Despite accumulation of substantial amounts of lutein and zeaxanthin, none of these carotenoids protects ARPE-19 cells from photo-oxidative damage caused by photosensitizers, which are present externally to the cells—either as water soluble, cell-impermeable rose bengal or lipophyllic all-*trans*-retinal incorporated into lipid vesicles. Based on the lack of protective effects of carotenoids supplemented over a period of 19 days against photo-oxidative damage induced by photosensitizers not penetrating through the plasma membrane, it can be suggested that lutein and zeaxanthin accumulate either inside the cell and/or are bound within the plasma membrane, preventing their action as singlet oxygen quenchers. Partial protection from photo-oxidative damage induced by extracellular photosensitizers is achieved by all three carotenoids, with each having similar efficiency, but only when they are present extracellularly together with the photosensitizers. Upon injection into PBS of carotenoids solubilized in DMSO, the carotenoids largely precipitate. At least some of the precipitates will likely accumulate in the plasma membrane where it can get re-solubilized. The protective effect of carotenoids administered in this manner suggests that this is the case. These findings underline the importance of co-localization of antioxidants with potential sources of reactive oxygen species. Some previous studies also indicate that zeaxanthin and/or lutein can exert protection in cultured ARPE-19 cells against photo-oxidative damage, and that protection is more efficient if carotenoids are co-localized with photosensitizers [47,115,116,117]. Wrona et al. have shown that zeaxanthin supplemented by ARPE-19 cells exposed to visible light and merocyanine 540, a well-characterized photosensitizer which associates with cell membranes, can substantially diminish oxidation of cholesterol but does not offer a significant protection against cytotoxicity unless it is given in combination with vitamin C or vitamin E and the exposure to photo-oxidation is short [47]. In those studies, zeaxanthin was administered to ARPE-19 cells 24 h prior to exposure to the photosensitizer and light. In other studies, where the photosensitizers were present inside the cells, such as phagocytosed lipofuscin, melanolipofuscin, or melanosomes, a combination of zeaxanthin with vitamin E were, at least partly, preserving cell viability [115,116,117]. In these experiments, pigment granules were enriched with antioxidants prior to administering to cultured cells. However, lutein and zeaxanthin do not accumulate in substantial concentrations in the human RPE [34,35].

The photoreceptor outer segments are the main layers in the retina where lutein and zeaxanthin are detected in substantial concentrations [34,35]. Potent photosensitizers can transiently accumulate in these parts of neurons, and, upon absorption of light, generate singlet oxygen [19,20,21,106]. In photoreceptor outer segments, all-*trans*-retinal is hydrolysed from photoactivated visual pigments, and its clearance is dependent mainly on activities of NADPH-dependent retinol dehydrogenase and the ATP-binding cassette transporter rim protein (ABCR, also known as ABCA4) [19,21,36,55,56,118]. The loss of function mutations in genes coding these enzymes result in delayed clearance of all-*trans*-retinal and increased susceptibility to light-induced damage to the retina in the mouse. In the human retina, mutations in the ABCA4 gene can lead to Stargardt’s disease, and are associated with increased risk of a certain subtype of retinitis pigmentosa and age-related macular degeneration. It is believed that photoexcitation of all-*trans*-retinal leads to the generation of a singlet oxygen and initiates the damage. The photoreceptor outer segments are rich in polyunsaturated fatty acids, such as docosahexaenoic acid, which are very susceptible to oxidation by singlet oxygen and subsequent formation of lipid hydroperoxides. Lipid hydroperoxides may themselves undergo decomposition in the presence of iron ions. Iron ions are known to accumulate with age in the retina and that accumulation is increased in age-related macular degeneration. Decomposition of lipid hydroperoxides generates lipid peroxyl radicals, which can propagate lipid peroxidation, resulting in the formation of end-products with photosensitizing properties [97]. Since these photosensitizing species are not distributed uniformly and are likely to be localized in lipophilic structures, it is important to determine to what extent carotenoids present in the photoreceptor outer segments are free and localized in the lipid membranes or are bound to proteins in the cytosolic environment. The localization of dehydrolutein present within the retina remains incompletely investigated. Its presence in the lipid membrane would be significant given its ability to function as a photoprotective agent.

This research addresses the role of non-pro-vitamin A carotenoids, especially the hydroxyl carotenoids that function intact within the retina where they shield the retina from oxidative damage that results from the activation of oxygen by blue light. The significance is such that lutein and zeaxanthin are components of the AREDS2 formula. The AREDS and AREDS2 clinical studies were separate 5-year studies carried out by the National Eye Institute, USA, and were large randomized, multi-centre, double-masked, placebo-controlled clinical trials, each recruiting >4000 AMD patients. These trials demonstrated the benefits of an antioxidant supplement during the 5-year study periods [32,119]. More recently, lutein and zeaxanthin have been identified within the brain and associated with a cognitive benefit and are currently under further investigation as treatments to slow the cognitive decline in aging and dementia [120,121,122,123]. As stated in the introduction, increasing macular pigment density has been demonstrated to improve several aspects of the visual function in addition to their photoprotection function [14,23,24,25,26,27,28,29,30,31]. Carotenoids are unsurpassed singlet oxygen quenchers and accumulate in the retina in remarkably high concentrations (>1 mM), thereby, having the potential to inhibit damage induced in the retina by blue light-generated singlet oxygen. Understanding the high specificity and selectivity of carotenoid uptake and accumulation mechanisms is essential for developing and improving formulations that deliver these carotenoids efficiently to the retina. An additional benefit of the study of carotenoids uptake and function in ARPE-19 cells is the potential that this research can provide novel insights into the delivery across the retina/brain-blood barrier that may open avenues for the delivery of therapeutics to retinal tissues and the treatment of a range of ocular pathologies.

## 5. Conclusions

In conclusion, our results demonstrate that dehydrolutein is a singlet oxygen quencher, which is similarly effective in protecting from photooxidative damage as lutein or zeaxnthin. Our findings point to the importance of determining whether dehydrolutein is present in the retina as a free carotenoid in lipid membranes or bound to proteins. In order to elucidate the role of RPE in selective carotenoid transport, further investigation of the mechanism of transport should be undertaken. This information will provide insight and enable further development of therapeutic strategies for increasing carotenoid accumulation in lipid membranes of the RPE and photoreceptor outer segment membrane where they are most needed as singlet oxygen quenchers and can most effectively protect against light-induced injuries. These injuries can contribute to the development of age-related macular degeneration and other diseases associated with increased (photo)oxidative stress and/or lutein/zeaxanthin abnormalities, namely Stargardt’s disease, retinitis pigmentosa, diabetic retinopathy, glaucoma, Sjögren-Larsson syndrome, and macular telangiectasia.

## Figures and Tables

**Figure 1 antioxidants-10-00753-f001:**
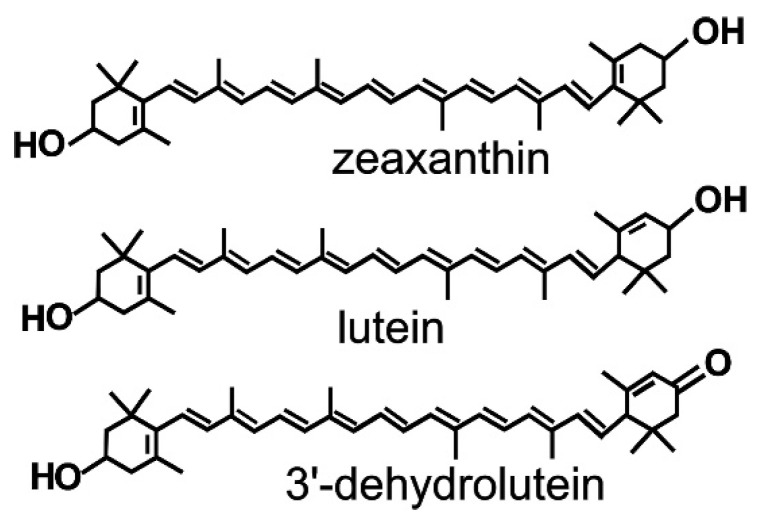
Structure of dietary carotenoids, zeaxanthin, and lutein, and their metabolite, dehydrolutein.

**Figure 2 antioxidants-10-00753-f002:**
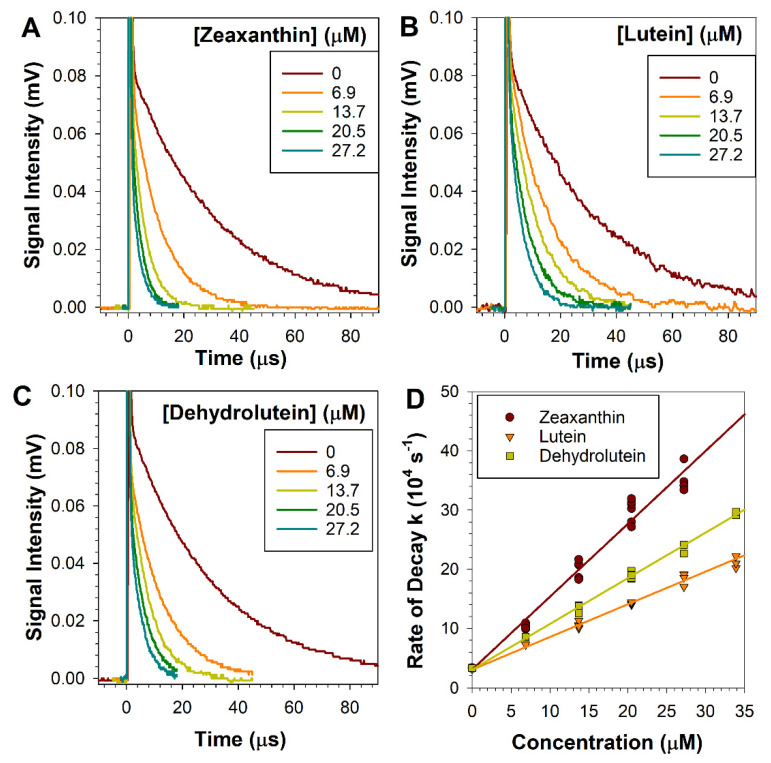
**Dehydrolutein quenches singlet oxygen with a similar efficiency to that of lutein or zeaxanthin**. (**A**–**C**): Representative kinetics of formation and decay of singlet oxygen after a 5-ns laser pulse photoexcitation of all-*trans*-retinal values in the absence and presence of dehydrolutein, lutein, and zeaxanthin at indicated concentrations. Time 0 indicates when the sample was exposed to the laser flash. The decays were fitted to exponential decay curves to obtain fitted parameters corresponding to the rates of decay. (**D**): Rates of singlet oxygen decay plotted as a function of carotenoid concentration. The quenching rate constants were determined by fitting straight lines to obtain the fitted slopes, which correspond to the rates of quenching.

**Figure 3 antioxidants-10-00753-f003:**
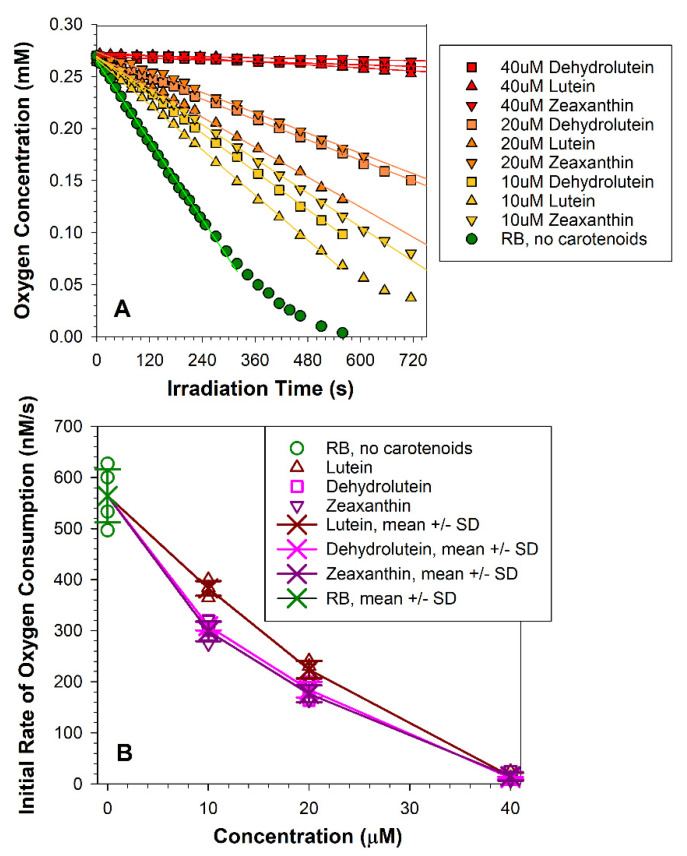
**Dehydrolutein inhibits rose-bengal-mediated photo-oxidation with similar efficiency to lutein and zeaxanthin**. (**A**) Representative kinetics of oxygen consumption during exposure to rose bengal and green light of liposomes in the absence and presence of carotenoids at indicated concentrations. Liposomes, consisting of 1 mg/mL egg yolk phosphatidylcholine (EYPC), 2.6 mM cholesterol, 11.8 mM 1,2-dimyristoyl-sn-glycero-3-phosphocholine (DMPC), and indicated concentration of carotenoids were exposed to green light (542 ± 4 nm, 14.0 mW/cm^2^) in the presence of 16-µM rose bengal (RB). The samples included 0.1 mM, 4-protio-3-carbamoyl-2,2,5,5-tetraperdeuteromethyl-3-pyrroline-1-yloxy (mHCTPO) used as a spin probe. The initial linear portions of the kinetics were fitted to straight lines, and the slopes of these lines gave initial rates of oxygen consumption. (**B**) The initial rates of oxygen consumption in the absence and presence of indicated carotenoids as a function of carotenoid concentration. Statistically significant differences between the initial rates of oxygen consumption in the presence of carotenoids used at the same concentration were only for 10 µM lutein, which was significantly faster than for 10 µM dehydrolutein or zeaxanthin (*p* = 0.005 and 0.002, respectively).

**Figure 4 antioxidants-10-00753-f004:**
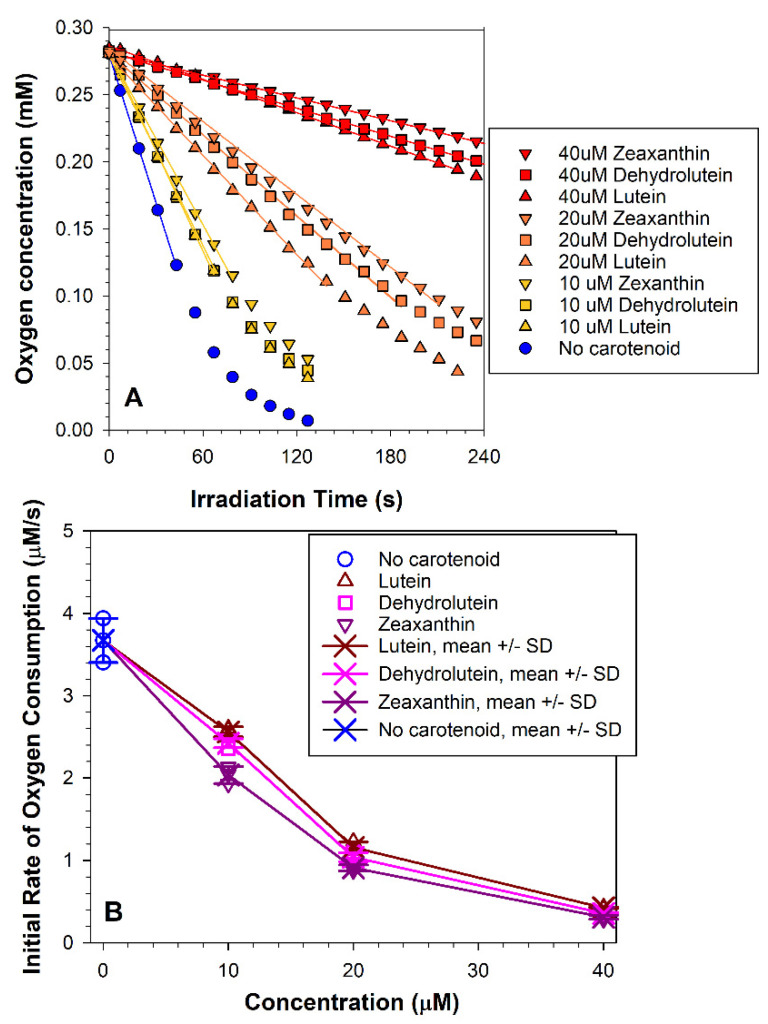
**Dehydrolutein inhibits all-*trans*-retinal-mediated photo-oxidation with similar efficiency to lutein and zeaxanthin**. (**A**) Representative kinetics of oxygen consumption during exposure of liposomes to all-*trans*-retinal and blue light in the absence and presence of carotenoids at indicated concentrations. Liposomes, consisting of 2.5 mg/mL EYPC, 0.75 mM all-*trans*-retinal, and indicated concentration of carotenoids, were exposed to blue light (404 ± 6 nm, 8.1 mW/cm^2^) in the presence of 1% DMSO. The samples included 0.1 mM mHCTPO used as a spin probe. The initial linear portions of the kinetics were fitted to straight lines, and the slopes of these lines gave the initial rates of oxygen consumption. (**B**) The initial rates of oxygen consumption in the absence and presence of indicated carotenoids as a function of carotenoid concentration. Statistically significant differences between the initial rates of oxygen consumption were observed between the carotenoid at 10 µM concentrations only. Oxidation in the presence of 10 µM zeaxanthin was significantly slower than for 10 µM dehydrolutein or lutein (*p* < 0.001, in both cases).

**Figure 5 antioxidants-10-00753-f005:**
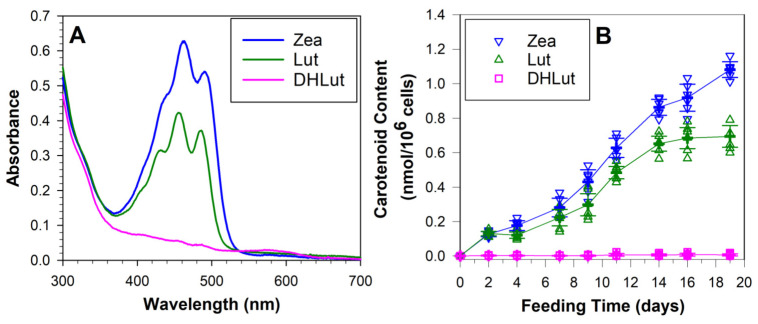
**ARPE-19 cells accumulate lutein and zeaxanthin but not dehydrolutein**. (**A**) Representative absorption spectra of extracts from ARPE-19 cells supplemented for 19 days with cell culture medium containing 10% FCS enriched with 2 µM lutein (Lut), dehydrolutein (DHLut), or zeaxanthin (Zea). (**B**) Concentrations of lutein, zeaxanthin, and dehydrolutein in ARPE-19 cells fed for indicated number of days with culture medium enriched with 2-µM carotenoids.

**Figure 6 antioxidants-10-00753-f006:**
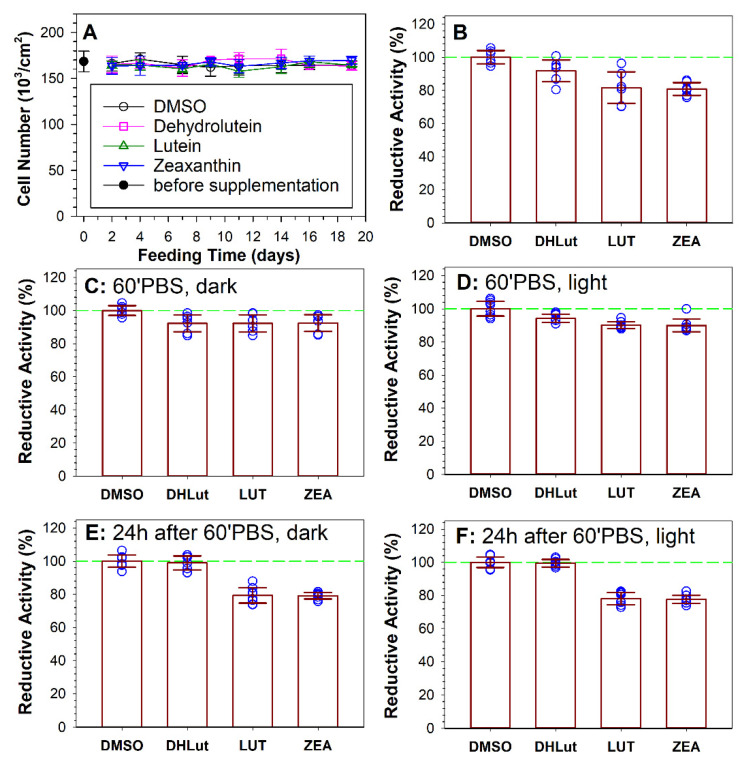
Supplementation with carotenoids does not affect cell viability but dehydrolutein can decrease their reductive activities. (**A**): ARPE-19 cell density before and after feeding with carotenoids for up to 19 days. The symbols represent means, while the error bars represent SDs. (**B**): Reductive activities of cells after 19 days of supplementation with vehicle (DMSO) or carotenoids: dehydrolutein (DHLut), lutein (LUT), and zeaxanthin (ZEA) measured by the MTT assay. (**C**,**D**): Reductive activities of cells after 19 days of supplementation with vehicle (DMSO) or indicated carotenoids measured by MTT assay after 1 h incubation in Dulbecco’s phosphate buffered saline (PBS) in dark (**C**) or with concomitant exposure to white light (**D**). (**E**,**F**): Reductive activities of cells after 19 days of supplementation with vehicle (DMSO) or indicated carotenoids measured by MTT assay 24 h after 1 h incubation in PBS in dark (**E**) or with exposure to white light (**F**). (**B**–**F**): the symbols correspond to individual measurements, the heights of the bars correspond to the means, and the error bars correspond to SDs.

**Figure 7 antioxidants-10-00753-f007:**
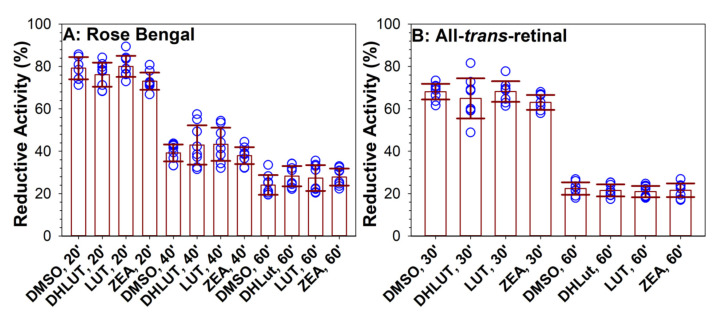
Supplementation with carotenoids for 19 days does not affect reductive activity of cells following damage induced by photosensitizers and light. Reductive activities of cells after 19 days of supplementation with vehicle (DMSO) or carotenoids: dehydrolutein (DHLut), lutein (LUT), and zeaxanthin (ZEA) measured by MTT assay immediately after 20, 40, or 60 min exposure to visible light (0.46 mW/cm^2^) and 0.5 μM rose bengal (A) or 24 h after 30 or 60 min exposure to visible light and liposomes containing 0.5 mM all-*trans*-retinal and 2.8 mg/mL EYPC (B). The reductive activities are expressed as ratios of reductive activity of cells in the presence of the photosensitizer to reductive activity of cells exposed to the same treatment in the absence of the photosensitizer. The symbols indicate individual measurements, the heights of the bars indicate the means, and the error bars indicate SDs. There were no statistically significant differences between DMSO and carotenoid-supplemented cells under each duration of exposure to light and rose bengal or all-*trans*-retinal.

**Figure 8 antioxidants-10-00753-f008:**
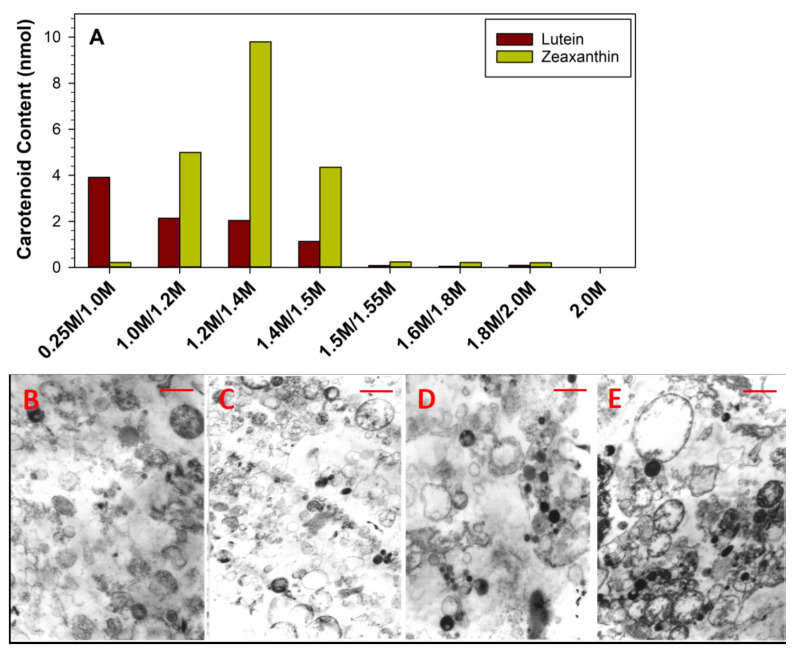
(**A**): Carotenoid content in subcellular fractions isolated on the discontinuous sucrose gradient at indicated interfaces of sucrose density. Representative TEM micrographs of organelles isolated from 0.25/1.00 M (**B**), 1.0/1.2 M (**C**), 1.2/1.4 M (**D**), and 1.4/1.5 M (**E**) sucrose. The length of the bar corresponds to 1 µm.

**Figure 9 antioxidants-10-00753-f009:**
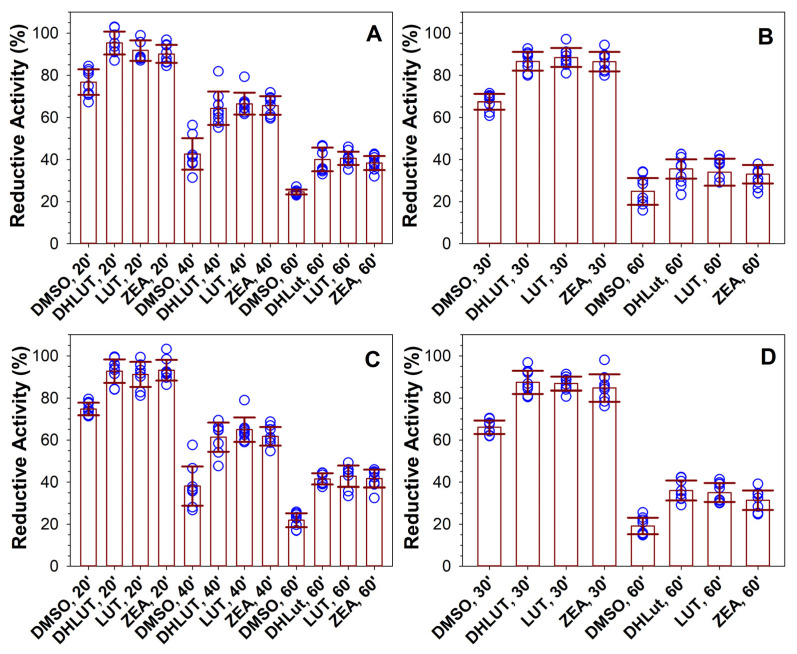
The presence of extracellular carotenoids partly protects cells from damage induced by photosensitizers and light independently of whether the cells underwent prior supplementation with carotenoids. (**A**,**B**): Reductive activities of ARPE-19 cells after 19 days of supplementation with vehicle (DMSO) or carotenoids: dehydrolutein (DHLut), lutein (LUT), and zeaxanthin (ZEA) measured by MTT assay immediately after 20, 40, or 60 min of exposure to visible light and 0.5 µM rose bengal (A) or 24 h after 30 or 60 min of exposure to visible light and liposomes containing 0.5 mM all-*trans*-retinal and 2.8 mg/mL EYPC (**B**). The exposure to photosensitizers and light was in the presence of 0.2% DMSO and 2 µM carotenoids solubilized in DPBS. (**C**,**D**): Reductive activities of cells without prior feeding with carotenoids, which were exposed to light, photosensitizers, and carotenoids. The MTT assay was performed immediately after the exposure to light and 0.5 µM rose bengal (**C**) or 24 h after the exposure to light and all-*trans*-retinal (**D**). The exposure to photosensitizers and light was in the presence of 0.2% DMSO and 2 µM carotenoids solubilized in DPBS. The symbols indicate individual measurements, the heights of the bars indicate means, and the error bars indicate SDs. Treatments with carotenoids resulted in a statistically significant increase in the reductive activities in comparison with DMSO-only-treated cells in all types of experiments (*p* < 0.001). The differences between the effects of different carotenoids in the same type of exposure were not statistically significant.

## Data Availability

The numerical data presented in the manuscript are available upon request from the corresponding author.

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
