# Peer review of "Comparison of Antioxidant Properties of Dehydrolutein with Lutein and Zeaxanthin, and their Effects on Cultured Retinal Pigment Epithelial Cells"

_antioxidants, 2021, doi:10.3390/antiox10050753_

Round 1
Reviewer 1 Report
This study compares the antioxidant properties of the carotenoids zeaxanthin, lutein and dehydrolutein. These carotenoids exist in relative high concentration in the primate retina where they may protect photoreceptors and the retinal pigment epithelium against light induced oxidative stress. A notable feature of this study is the elegant combination of in vitro and in vivo studies, which provides a thorough analysis of putative neuroprotective effects of these pigments. In general, the article is well written and most of the conclusions are justified by the experimental data. Major findings are that dehydrolutein displays similar antioxidant properties in the test tube as other macula pigments. This reviewer asks for a few corrections and has some suggestions that will further enhance the work.
Introduction: A figure displaying the chemical structures of zeaxanthin, lutein and dehydrolutein would be beneficial for the general readership of the paper.
Page 3, line ‘The absorbance of all-trans-retinal in a 1 cm square cuvette was about 0.17 at the excitation wavelength of 355 nm for all 112 the solutions’. At which concentration?
Page 3, line 174: Did the authors perform HPLC analysis of carotenoids to study whether the compounds undergo metabolic transformation in ARPE-19 cells? Do different carotenoids accumulate in the same cellular compartments?
Results:
Page 5, line 229: The involvement of carotenoid binding is very speculative. This statement would fit better into the discussion part. Binding proteins for carotenoids would need to exist in similar quantities as carotenoids which is definitively not the case in the primate retina. Moreover, GSTP1 is a small (~24 kDa) globular protein with a single glutathione-binding site located within the thioredoxin-like domain. Thus, the proposal that GSTP1 accommodates a large planar ligands such as zeaxanthins is very problematic. Even more so at the proposed carotenoid/protein stoichiometry of 2:1. For a recent discussion see https://doi.org/10.1016/j.bbalip.2019.158571.
Page 8, line 288: What are ‘light levels’?
Page 10, second paragraph: What is the concentration of lutein and zeaxanthin in the ARPE-19? Does it exceed the concentration on the cell culture medium? Do the carotenoids accumulate in the cells or are the associated with the plasma membrane? This information is critical to understand their putative beneficial and/or detrimental effects (see also below).
Figure 4: Can the authors exclude that dehydrolutein is selectively metabolized by the cells? Do ARPE-19 cells express enzymes that catabolize carotenoids?
Figure 5: The conclusion that carotenoid accumulation does not affect cell viability is surprising. The MTT assay is a cell viability and proliferation assay and decreased ‘reductive activity’ (MTT reduction) is indicative for this (even if it is only 20%). Additionally, this reviewer has the feeling that the results of the MTT assay are over-interpreted and too speculative. Instead, the authors should determine activity of glycolytic enzymes and respiration rate of carotenoid-treated cells. At least, this paragraph needs some rewording.
Figure 6: Is this effect dose-dependent. Would carotenoids protect cells against lower doses of the stressors? Is this an additive effect because data in figure 5 indicate that carotenoids already reduced the cells viability?
Figure 7: Same is true as in Figure 6. Dose-dependency of the effect needs to be demonstrated. Is the improved viability of cells in the short term experiments caused by the absence of detrimental effects as observed in the long term carotenoid-feeding experiment (see figure 5)?
Author Response
Response to Reviewer 1
We thank the Reviewer for the constructive criticism and prompts how to enrich the Discussion.
This study compares the antioxidant properties of the carotenoids zeaxanthin, lutein and dehydrolutein. These carotenoids exist in relative high concentration in the primate retina where they may protect photoreceptors and the retinal pigment epithelium against light induced oxidative stress. A notable feature of this study is the elegant combination of in vitro and in vivo studies, which provides a thorough analysis of putative neuroprotective effects of these pigments. In general, the article is well written and most of the conclusions are justified by the experimental data. Major findings are that dehydrolutein displays similar antioxidant properties in the test tube as other macula pigments. This reviewer asks for a few corrections and has some suggestions that will further enhance the work.
Introduction: A figure displaying the chemical structures of zeaxanthin, lutein and dehydrolutein would be beneficial for the general readership of the paper.
The figure with structures has been added as Figure 1.
Page 3, line ‘The absorbance of all-trans-retinal in a 1 cm square cuvette was about 0.17 at the excitation wavelength of 355 nm for all 112 the solutions’. At which concentration?
The concentration of 5.3 µM was added to the Methods (line 118).
Page 3, line 174: Did the authors perform HPLC analysis of carotenoids to study whether the compounds undergo metabolic transformation in ARPE-19 cells? Do different carotenoids accumulate in the same cellular compartments?
For the dehydrolutein-supplemented cells, we have not performed HPLC analysis because we could not detect ANY carotenoids or apocarotenoids by absorption spectroscopy, whereas the absorption spectra of lutein or zeaxanthin-supplemented cells did not indicate any changes (blue shift of the absorbance maximum) suggesting the formation of apocarotenoids. The accumulation of lutein and zeaxanthin could be easily seen with naked eye after washing wells with PBS when the culture plate was placed on white background – the wells with cells supplemented with lutein or zeaxanthin appeared orange whereas wells with cells supplemented with only DMSO or dehydrolutein remained colourless.
We do not know whether lutein, zeaxanthin, and dehydrolutein accumulate in the same cellular compartments. It is well documented that lutein and zeaxanthin accumulate in the retina in different areas as a function of eccentricity from the fovea centre, with zeaxanthin being highly concentrated in the fovea while lutein is distributed more diffusely across the retina [1-3]. It has been determined using confocal resonance Raman microscopy that zeaxanthin:lutein ratio can be greater than 9:1 in the centre of the fovea, decreases to 4:1 200 µm from that centre, and 200 µm further away the ratio becomes 1:4 [3]. In the peripheral retina, >5 mm from the foveal center, the total concentration of zeaxanthin+lutein is approximately 1/100 that of the central fovea [4]. Across the retinal layers in the fovea, the greatest concentrations of each, zeaxanthin and lutein, are in the Henle’s fibre layer and are clearly visible in microsections [5]. The Henle fiber layer contains axons of foveal photoreceptors, outer plexiform layer where axonal terminals of photoreceptors make synaptic connections with bipolar and horizontal cell, inner nuclear layer with cell bodies of glial Mueller cells and three types of neurons: bipolar, horizontal and amacrine cells, inner plexiform layer with synaptic connections between bipolar cells with amacrine and retinal ganglion cell, and outer nuclear layer with cell bodies of photoreceptors. The Mueller cell processes spread across all these layers and there is growing body of evidence that macular carotenoids can be bound also to these glial cells (reviewed by Curcio [6]).
It has been demonstrated that the majority of lutein and zeaxanthin in the macula is highly organized and concentrically aligned relative to the central axis passing through the pupil and the macula, i.e. the visual axis [7]. Such an organizationa feature of the macular pigment is the accepted explanation of the entoptical phenomenon known as Haidinger's brushes. Haidinger’s brushes best can be seen as an hour-glass shaped darkening when horizontally plane-polarized short wavelength (blue) light passes through the retina at the center of the visual field and is attributed to preference of the concentrically arrayed pigment molecules laying above and below the axis to absorb the horizontally plane polarized light whereas molecules laying either to the left or right of the axis do not. This hour-glass figure entopical figure rotates with the plane of the polarized light, always being oriented at 90° to the plane of polarization, and has an action spectrum identical to that of the carotenoids, lutein and zeaxanthin which compose the macular pigment [7]. The arrangement of lutein and zeaxanthin arises because Henle’s fibres extend radially from the center of the fovea outward to the peripheral retina and the carotenoids are either incorporated spanning the bilayer lipid membrane or are associated with proteins that are axially arrayed along axon length resulting in a similar organizational arrangement of the carotenoids.
It has been shown that carotenoids can bind to several proteins [8]. They include tubulin, which is abundant in axons, but the binding is of low specificity and affinity. Also, the interphotoreceptor retinoid binding protein (IRBP), which is present in the space around inner and outer segments of photoreceptors, binds both lutein and zeaxanthin with low specificity, but the affinity is high, similar to that of retinoids [9]. These are constituents of all species and so are not felt to be the primary candidates responsible for the binding of carotenoids of the macular pigment, a feature that is anatomical unique to the primate retina. In addition to those mentioned above, there are other proteins in the primate retina that bind to carotenoids with high specificity and affinity: steroidogenic acute regulatory domain 3 (StARD3) and a pi isoform of glutathione S-transferase (GSTP1), which bind to lutein and zeaxanthin, respectively [8].
GSTP1 is considered to be a cytosolic enzyme whereas the majority of the lutein and zeaxanthin in the retina are membrane-associated and require detergent to solubilize them [10,11]. However, it cannot be excluded that GSTP1 undergoes posttranslational modification which anchors it in the lipid membrane. It has been reported that the immunofluorescence of antibody to human recombinant GSTP1 appears throughout all retinal layers in the human macula, including inner and outer segments of photoreceptors, with the greatest expression in the inner and outer plexiform layers, where retinal neurons make synaptic junctions [11]. It appears absent in the nuclei of retinal cells. In another report, the strongest labelling in the monkey macula appears along the outer limiting membrane with some cone inner segments stained heavily while others not at all [8]. It has been proposed that GSTP1 can bind zeaxanthin at the carotenoid to protein stoi-chiometry of 2:1 [11]. It has been pointed out that GSTP1 is a small, 24 kDa globular protein so binding a long planar ligands such as zeaxanthin could expose a large portion of that lipophilic ligand to the external environment [12].
StARD3 was detected by Western blotting in the human neural retina and RPE-choroid, and its mRNA expression was confirmed by RT-PCR in the human retina [13]. Immunocytochemistry on the monkey retina demonstrated StARD3 antibody binding in retinal layers spanning from ganglion cell layer to the outer limiting membrane with the greatest density in the Henle’s fibre layer containing photoreceptor axons [8,13]. Also RPE was heavily stained but that staining was non-specific to the StARD3 antibody. The retinal layers of the outer segments of photoreceptors containing membraneous discs packed with visual pigments, and inner segments of photoreceptors with mitochondria, endoplasmic reticulum and Golgi apparatus, as well as interphotoreceptor matrix filling the space between photoreceptors exhibited relatively low staining intensity [13]. Co-labelling cone photoreceptors showed that StARD3 antibody labels cones with greater intensity than rods, whereas co-labelling of Müller cells showed no co-localization of the StARD3 antibody with Müller cells.
It has been demonstrated that optical density of the macular pigments can be increased by supplementation with lutein over a period of 140 days [14]. Once the supplementation was stopped, the increase in macular pigment density continued for another 43 days and reached a stable level which remained the same for at least the following 150-200 days when the monitoring was stopped. The long-term stability of macular pigment in the retina over this period after discontinuation of supplementation is remarkable considering the environment in the retina which is exposed daily to high fluxes of light in the presence of various photosensitizers and high abundance of polyunsaturated lipids. Carotenoids are rather labile compounds susceptible to oxidative degradation. Therefore, their stability in the macula also suggest that they are protected from oxidation and degradation by proteins and/or sequestered in other immobilizing structures.
It remains an open question which subcellular compartments accumulate lutein and zeaxanthin, and whether or not they accumulate in the same structures, i.e. cytoplasm, endosomes, lysosomes, phagosomes, phagolysosomes, mitochondria, endoplasmic reticulum membranes, or subcellular protein framework although as noted earlier evidence supports cell lipid bilayers or other radially arrayed structures within the retina. To investigate the localization in ARPE-19 cell, we performed subcellular fractionation using differential centrifugation. Most lutein and zeaxanthin were lost in the pellet from low-speed centrifugation at 60 g and/or in the supernatant from pelleting the post-nuclear fraction. The postnuclear fraction was separated by ultracentrifugation on a discontinuous sucrose gradient. Lutein and zeaxanthin were detectable at interfaces at several sucrose densities starting at 0.25/1.0 M down to 1.4 M. Examination of these bands by transmission electron microscopy revealed that these fractions were rich mostly in endosomes, lysosomes, phagolysosomes, and some mitochondria. We have added Subcellular fractionation and transmission electron microscopy (TEM) section to the Methods (lines 216-236), included a figure showing the results (Fig. 8) and included results description (lines 484-499), and the discussion (lines 675-688) in the revised manuscript.
Results:
Page 5, line 229: The involvement of carotenoid binding is very speculative. This statement would fit better into the discussion part. Binding proteins for carotenoids would need to exist in similar quantities as carotenoids which is definitively not the case in the primate retina. Moreover, GSTP1 is a small (~24 kDa) globular protein with a single glutathione-binding site located within the thioredoxin-like domain. Thus, the proposal that GSTP1 accommodates a large planar ligands such as zeaxanthins is very problematic. Even more so at the proposed carotenoid/protein stoichiometry of 2:1. For a recent discussion see https://doi.org/10.1016/j.bbalip.2019.158571.
As suggested, we have moved the part to the Discussion, where we have also expanded the considerations of the carotenoid-binding proteins (lines 574-582 and 587-688).
Page 8, line 288: What are ‘light levels’?
Thank you to the Reviewer for spotting this typographical error. It has been corrected to “high levels.”
Page 10, second paragraph: What is the concentration of lutein and zeaxanthin in the ARPE-19? Does it exceed the concentration on the cell culture medium? Do the carotenoids accumulate in the cells or are the associated with the plasma membrane? This information is critical to understand their putative beneficial and/or detrimental effects (see also below).
We have provided the carotenoid concentrations in Fig. 5B expressed in nmol/mln cells. Based on these data, the average millimolar concentrations in cells can be estimated. Assuming the height of the cell monolayer of 8 µm, and the 25 cm2 surface area of the cell culture flask, the 4.2 million cells in that flask occupied a volume of 0.02 ml, giving the average concentration of 0.23 and 0.15 mM for zeaxanthin and lutein, respectively, after 19 days of supplementation. This means that cells accumulated 116- and 74-fold greater concentrations of carotenoids than those provided in the culture medium. This estimation was added to the 2nd paragraph of section 3.3, lines 374-379.
Based on the lack of protective effects of carotenoids supplemented over the period of 19 days against photooxidative damage induced by photosensitizers not penetrating through the plasma membrane, it can be suggested that lutein and zeaxanthin accumulate either inside the cell and/or are bound within plasma membrane preventing their action as singlet oxygen quenchers. This consideration is included in the Discussion in lines 750-755.
Upon injection into PBS of carotenoids solubilized in DMSO, the carotenoids largely precipitate. At least some of the precipitates will likely accumulate in the plasma membrane where it can get re-solubilized. The protective effect of carotenoids administered in this manner suggests that this is the case. This consideration was added to the Discussion in lines 757-760.
Figure 4: Can the authors exclude that dehydrolutein is selectively metabolized by the cells? Do ARPE-19 cells express enzymes that catabolize carotenoids?
No, we cannot exclude that possibility. It has been shown by Thomas and colleagues that β-carotene-9’,10’-dioxygenase (BCO2) which catabolizes various carotenoids and apo-carotenoids, can convert dehydrolutein into apocarotenoids [15]. It has been reported that ARPE-19 cells express this enzyme using qRT-PCR and Western blotting and that exposure to lutein further increases BCO2 mRNA [16].
The experimental difficulty is that the commercial antibody used to detect BCO2 in the human and mouse retinas by Li and colleagues [17] was shown later to be non-specific to BCO2 and labelled something else in both bcdo2+/+ and bcdo2-/- mice [15]. We have expanded the Discussion considering a possibility that lutein and zeaxanthin may avoid enzymatic degradation by binding to proteins whereas dehydrolutein remains unbound and is an available substrate for that enzyme. Another pathway by which ARPE-19 cells could remove dehydrolutein is by incorporatiing it into lipoproteins they synthesize and secrete. Curcio and colleagues demonstrated RPE cells can synthesize and secrete lipoproteins [6,18], while Amin et al. [19] demonstrated that ARPE-19 cells can secrete membranous sub-RPE deposits when supplemented with retinal extract . During and colleagues demonstrated that CaCo-2 cells can synthesize and secrete carotenoid-containing lipoproteins [20,21]. We have added these considerations into the Discussion in lines 707-722.
Figure 5: The conclusion that carotenoid accumulation does not affect cell viability is surprising. The MTT assay is a cell viability and proliferation assay and decreased ‘reductive activity’ (MTT reduction) is indicative for this (even if it is only 20%). Additionally, this reviewer has the feeling that the results of the MTT assay are over-interpreted and too speculative. Instead, the authors should determine activity of glycolytic enzymes and respiration rate of carotenoid-treated cells. At least, this paragraph needs some rewording.
The results presented in our manuscript show that the decrease in the reductive activity of cells with accumulated lutein and zeaxanthin is reversible: after depriving cells from nutrients present in the culture medium for an hour, the reductive activity increased to the levels similar to cells without accumulated carotenoids once the glucose and other nutrients became available. MTT assay is widely used as a measure of cell viability but, as stated in the manuscript, what it measures is the reduction of MTT to formazan which can be affected by activity of various enzymes and availability of NAD(P)H [22]. It was demonstrated that MTT reductive activity is not affected, or in some cases even enhanced (by sparing NADH), by inhibitors of mitochondrial electron transport chain, but it decreases in the presence of inhibitors of glucose uptake and cytoplasmic glycolysis [23]. Therefore, it has been suggested that MTT reduction can be viewed as a measure of the glycolytic activity and NAD(P)H production [23]. It has been shown that lutein supplementation can affect activities of several enzymes in the heart and kidney a rat model of diabetes [24], can modulate expression of a number of genes in murine liver [25], and in ARPE-19 cells activates the transcription factor, nuclear factor erythroid 2-related factor 2 (Nrf2) which, in turn, activates genes encoding antioxidative and phase II enzymes that are involved in the maintenance of the cellular redox status as well as in the detoxification of xenobiotics, including NAD(P)H:quinone oxidase, NQO1 [26]. Therefore, the decreased reductive activity in the presence of lutein and zeaxanthin may reflect their effects on NAD(P)H and glycolysis. Moreover, in addition to the MTT assay, we used cell counting as a measure of cell viability. As shown in Fig. 6A, the cell numbers remained similar to the initial ones throughout the duration of the supplementation. We have added these considerations to the Discussion. We have expanded the Discussion by including there these considerations in lines 723-745.
We agree that the effects of lutein and zeaxanthin accumulation on reductive activity of cells towards MTT is intriguing and deserves further investigation which could include measurements of the rates of acidification and oxygen consumption, as well as the levels of ATP.
Figure 6: Is this effect dose-dependent. Would carotenoids protect cells against lower doses of the stressors? Is this an additive effect because data in figure 5 indicate that carotenoids already reduced the cells viability?
We performed the experiments also at exposure times to light of 20 and 40 minutes for rose bengal and for 30 minutes for all-trans-retinal. No protective effects could be seen at these shorter exposure times, but the reductive activities were more variable under each of the conditions than for 1 h exposures. We have added these data to Fig. 7.
As stated in in the Results (lines 441-446) and Methods (lines 202-204), the reductive activity was calculated by dividing the absorbance of formazan in cells treated with carotenoids and photosensitizers by absorbance of formazan formed by cells from the same plate treated with the corresponding carotenoids and exposed to light in the absence of the photosensitizer.
Figure 7: Same is true as in Figure 6. Dose-dependency of the effect needs to be demonstrated. Is the improved viability of cells in the short term experiments caused by the absence of detrimental effects as observed in the long term carotenoid-feeding experiment (see figure 5)?
We performed the experiments also at exposure times to light of 20 and 40 minutes for rose bengal and for 30 minutes for all-trans-retinal. We have added these data to Fig. 9.
References
- Bone, R.A.; Landrum, J.T.; Friedes, L.M.; Gomez, C.M.; Kilburn, M.D.; Menendez, E.; Vidal, I.; Wang, W. Distribution of lutein and zeaxanthin stereoisomers in the human retina. Exp. Eye Res. 1997, 64, 211-218, doi:10.1006/exer.1996.0210.
- Landrum, J.T.; Bone, R.A. Lutein, zeaxanthin, and the macular pigment. Arch. Biochem. Biophys. 2001, 385, 28-40, doi:10.1006/abbi.2000.2171.
- Li, B.X.; George, E.W.; Rognon, G.T.; Gorusupudi, A.; Ranganathan, A.; Chang, F.Y.; Shi, L.J.; Frederick, J.M.; Bernstein, P.S. Imaging lutein and zeaxanthin in the human retina with confocal resonance Raman microscopy. Proc. Natl. Acad. Sci. U. S. A. 2020, 117, 12352-12358, doi:10.1073/pnas.1922793117.
- Bone, R.A.; Landrum, J.T.; Fernandez, L.; Tarsis, S.L. Analysis of the macular pigment by HPLC: retinal distribution and age study. Invest. Ophthalmol. Vis. Sci. 1988, 29, 843-849.
- Snodderly, D.M.; Auran, J.D.; Delori, F.C. The macular pigment. II. Spatial distribution in primate retinas. Invest. Ophthalmol. Vis. Sci. 1984, 25, 674-685.
- Curcio, C.A. Antecedents of Soft Drusen, the Specific Deposits of Age-Related Macular Degeneration, in the Biology of Human Macula. Invest. Ophthalmol. Vis. Sci. 2018, 59, AMD182-AMD194, doi:10.1167/iovs.18-24883.
- Bone, R.A.; Landrum, J.T. Macular pigment in Henle fiber membranes: a model for Haidinger's brushes. Vision Res. 1984, 24, 103-108, doi:10.1016/0042-6989(84)90094-4.
- Bernstein, P.S.; Li, B.X.; Vachali, P.P.; Gorusupudi, A.; Shyam, R.; Henriksen, B.S.; Nolan, J.M. Lutein, zeaxanthin, and meso-zeaxanthin: The basic and clinical science underlying carotenoid-based nutritional interventions against ocular disease. Prog. Retin. Eye Res. 2016, 50, 34-66, doi:10.1016/j.preteyeres.2015.10.003.
- Vachali, P.P.; Besch, B.M.; Gonzalez-Fernandez, F.; Bernstein, P.S. Carotenoids as possible interphotoreceptor retinoid-binding protein (IRBP) ligands: a surface plasmon resonance (SPR) based study. Arch. Biochem. Biophys. 2013, 539, 181-186, doi:10.1016/j.abb.2013.07.008.
- Sommerburg, O.G.; Siems, W.G.; Hurst, J.S.; Lewis, J.W.; Kliger, D.S.; van Kuijk, F.J. Lutein and zeaxanthin are associated with photoreceptors in the human retina. Curr. Eye Res. 1999, 19, 491-495, doi:10.1076/ceyr.19.6.491.5276.
- Bhosale, P.; Larson, A.J.; Frederick, J.M.; Southwick, K.; Thulin, C.D.; Bernstein, P.S. Identification and characterization of a Pi isoform of glutathione S-transferase (GSTP1) as a zeaxanthin-binding protein in the macula of the human eye. J. Biol. Chem. 2004, 279, 49447-49454, doi:10.1074/jbc.M405334200.
- Widjaja-Adhi, M.A.K.; Golczak, M. The molecular aspects of absorption and metabolism of carotenoids and retinoids in vertebrates. Biochim. Biophys. Acta Mol. Cell Biol. Lipids 2020, 1865, doi:10.1016/j.bbalip.2019.158571.
- Li, B.; Vachali, P.; Frederick, J.M.; Bernstein, P.S. Identification of StARD3 as a lutein-binding protein in the macula of the primate retina. Biochemistry 2011, 50, 2541-2549, doi:10.1021/bi101906y.
- Landrum, J.T.; Bone, R.A.; Joa, H.; Kilburn, M.D.; Moore, L.L.; Sprague, K.E. A one year study of the macular pigment: the effect of 140 days of a lutein supplement. Exp Eye Res 1997, 65, 57-62, doi:10.1006/exer.1997.0309.
- Thomas, L.D.; Bandara, S.; Parmar, V.M.; Srinivasagan, R.; Khadka, N.; Golczak, M.; Kiser, P.D.; von Lintig, J. The human mitochondrial enzyme BCO2 exhibits catalytic activity toward carotenoids and apocarotenoids. J. Biol. Chem. 2020, 295, 15553-15565, doi:10.1074/jbc.RA120.015515.
- Gong, X.; Draper, C.S.; Allison, G.S.; Marisiddaiah, R.; Rubin, L.P. Effects of the Macular Carotenoid Lutein in Human Retinal Pigment Epithelial Cells. Antioxidants (Basel) 2017, 6, doi:10.3390/antiox6040100.
- Li, B.X.; Vachali, P.P.; Gorusupudi, A.; Shen, Z.Q.; Sharifzadeh, H.; Besch, B.M.; Nelson, K.; Horvath, M.M.; Frederick, J.M.; Baehr, W., et al. Inactivity of human beta,beta-carotene-9 ', 10 '-dioxygenase (BCO2) underlies retinal accumulation of the human macular carotenoid pigment. Proc. Natl. Acad. Sci. U. S. A. 2014, 111, 10173-10178, doi:10.1073/pnas.1402526111.
- Curcio, C.A. Soft Drusen in Age-Related Macular Degeneration: Biology and Targeting Via the Oil Spill Strategies. Invest Ophthalmol Vis Sci 2018, 59, AMD160-AMD181, doi:10.1167/iovs.18-24882.
- Amin, S.; Chong, N.H.; Bailey, T.A.; Zhang, J.; Knupp, C.; Cheetham, M.E.; Greenwood, J.; Luthert, P.J. Modulation of Sub-RPE deposits in vitro: a potential model for age-related macular degeneration. Invest. Ophthalmol. Vis. Sci. 2004, 45, 1281-1288, doi:10.1167/iovs.03-0671.
- During, A.; Doraiswamy, S.; Harrison, E.H. Xanthophylls are preferentially taken up compared with beta-carotene by retinal cells via a SRBI-dependent mechanism. J. Lipid Res. 2008, 49, 1715-1724, doi:10.1194/jlr.M700580-JLR200.
- During, A.; Hussain, M.M.; Morel, D.W.; Harrison, E.H. Carotenoid uptake and secretion by CaCo-2 cells: beta-carotene isomer selectivity and carotenoid interactions. J. Lipid Res. 2002, 43, 1086-1095, doi:10.1194/jlr.m200068-jlr200.
- Stockert, J.C.; Horobin, R.W.; Colombo, L.L.; Blazquez-Castro, A. Tetrazolium salts and formazan products in Cell Biology: Viability assessment, fluorescence imaging, and labeling perspectives. Acta Histochem. 2018, 120, 159-167, doi:10.1016/j.acthis.2018.02.005.
- Berridge, M.V.; Herst, P.M.; Tan, A.S. Tetrazolium dyes as tools in cell biology: new insights into their cellular reduction. Biotechnol. Annu. Rev. 2005, 11, 127-152, doi:10.1016/S1387-2656(05)11004-7.
- Sharavana, G.; Joseph, G.S.; Baskaran, V. Lutein attenuates oxidative stress markers and ameliorates glucose homeostasis through polyol pathway in heart and kidney of STZ-induced hyperglycemic rat model. Eur. J. Nutr. 2017, 56, 2475-2485, doi:10.1007/s00394-016-1283-0.
- Serpeloni, J.M.; Colus, I.M.; de Oliveira, F.S.; Aissa, A.F.; Mercadante, A.Z.; Bianchi, M.L.; Antunes, L.M. Diet carotenoid lutein modulates the expression of genes related to oxygen transporters and decreases DNA damage and oxidative stress in mice. Food Chem. Toxicol. 2014, 70, 205-213, doi:10.1016/j.fct.2014.05.018.
- Frede, K.; Ebert, F.; Kipp, A.P.; Schwerdtle, T.; Baldermann, S. Lutein activates the transcription factor Nrf2 in human retinal pigment epithelial cells. J. Agric. Food Chem. 2017, 65, 5944-5952, doi:10.1021/acs.jafc.7b01929.
Reviewer 2 Report
This s a well-written paper. The main conclusion is that antioxidant properties of dehydrolutein are similar to those of lutein and zeaxanthin. Moreover, long-term exposure of ARPE-19 cells to dehydrolutein, lutein, or zeaxanthin does not affect cell viability, leads to asubstantial accumulation of lutein and zeaxanthin within the cells, but dehydrolutein does not accumulate to a significant extent. There is an extensive list of references in the field.
Major issues:
Why did you choose dehydrolutein for investigation? It should be explained in the introduction. Why ARPE-19 cells have beend used for this study? What is the main hypothesis for this study?
There are numerous results but discussion chapter is quite short.
Minor remarks:
Line 23: In the abstract the abbreviation ARPE-19 should be explained.
Line 512 it is written „memvranes” instead of ; I suppose: „membranes”
Line 334: there is no explanation for „SR-BI and LDL”
In the title it should be mentioned that cel cultures have been used.
Author Response
Response to Reviewer 2:
This s a well-written paper. The main conclusion is that antioxidant properties of dehydrolutein are similar to those of lutein and zeaxanthin. Moreover, long-term exposure of ARPE-19 cells to dehydrolutein, lutein, or zeaxanthin does not affect cell viability, leads to asubstantial accumulation of lutein and zeaxanthin within the cells, but dehydrolutein does not accumulate to a significant extent. There is an extensive list of references in the field.
Major issues:
Why did you choose dehydrolutein for investigation? It should be explained in the introduction. Why ARPE-19 cells have beend used for this study? What is the main hypothesis for this study?
The Introduction notes that high concentrations dehydrolutein accumulate in the human retina as a product of metabolic transformation of lutein or zeaxanthin, and the dehydrolutein content increases with age. While the antioxidant properties of lutein and zeaxanthin have been investigated extensively, no studies on antioxidant properties dehydrolutein have been reported. It is the consensus that the antioxidant properties of lutein and zeaxanthin contribute to maintaining the retina healthy by preventing photooxidative damage to sensitive retina structures and decreases the risk of AMD. Therefore, the antioxidant properties and potential role of dehydrolutein in protecting the retina from photosensitized oxidation is a topic that may provide new and significant insight relevant to human ocular health. The structural similarities of dehydrolutein to lutein, zeaxanthin as well as other carotenoids which are efficient singlet oxygen quenchers, prompted us to hypothesize that dehydrolutein may quench singlet oxygen and protect the retina from photooxidative damage. The retinal cells most susceptible to photodamage are the photoreceptors and retinal pigment epithelial cells. In culture, photoreceptors loose very rapidly their outer segments where visual pigments are present in high concentrations. This makes them inappropriate as an in vitro model of retina visual function but they have been found useful for biochemical and short-term physiological studies of the human retina. Dunn et al. reported in 1996 development of ARPE-19 cell line with many structural and functional properties characteristic of RPE cells in vivo and suggested that this cell line would be valuable for in vitro studies of retinal pigment epithelium physiology. Since then numerous groups have employed ARPE-19 cells; current PubMed search with a keyword “ARPE-19” gives 1893 papers. ARPE-19 cells are well characterized cell line derived from spontaneously raised RPE cell line of 19 year old male cadaver retaining several characteristics typical for the RPE such as very active phagocytosis and expression of receptors involved in carotenoid uptake and catabolism, such as SR-B1, LDLR, ABCA1, CD36 and BCO2. Unlike the primary cultures of human RPE which tend to lose rapidly the epithelial characteristics, ARPE-19 cells retain these characteristics for many passages.
ARPE-19 are used here as an in vitro model of the physiological situation with respect to transport of the carotenoids into the cells and across the blood retina barrier. Moreover, they are a useful cell line to demonstrate the comparative protective capabilities of the carotenoids studied as antioxidants and photoprotective agents to singlet oxygen produced externally to the plasma membrane. By employing confluent ARPE-19 cells forming a monolayer and lipid vesicles with all-trans-retinal, we mimic a physiological situation where all-trans-retinal accumulates in photoreceptor outer segments, which face the apical portion of the RPE. All-trans-retinal is a potent photosensitizer which generates singlet oxygen when photoexcited with blue light; singlet oxygen is a reactive oxygen species which can cause damage to unsaturated lipids and proteins.
The main hypothesis of this study was that dehydrolutein can quench singlet oxygen and protect from photooxidative damage where singlet oxygen is involved and should have a similar efficiency as lutein and zeaxanthin. The massive accumulation of lutein and zeaxanthin contrasts the the complete absence of transport of dehydrolutein by ARPE-19 cells was an unexpected finding. The transport of non-vitamin A carotenoids is only partially understood and this result provides a new path to understanding the processes that result in selective and highly specific accumulation of these potent antioxidants.
There are numerous results but discussion chapter is quite short.
We have moved a part of discussion which was included in the Results to the Discussion; moreover, we have considerably expanded the Discussion. The new parts of the Discussion are highlighted in yellow.
Minor remarks:
Line 23: In the abstract the abbreviation ARPE-19 should be explained.
ARPE-19 is not an abbreviation. It is a name given to that particular cell line derived from cadaver RPE from a 19-year old donor. The “A” in the name stands for Amy Aotaki-Keen who derived these cells.
Line 512 it is written „memvranes” instead of ; I suppose: „membranes”
Corrected.
Line 334: there is no explanation for „SR-BI and LDL”
The full names are now included before the abbreviations are used for the first time.
In the title it should be mentioned that cel cultures have been used.
The title has been changed as suggested so it reads: “Comparison of Antioxidant Properties of Dehydrolutein with Lutein and Zeaxanthin, and their Effects on Cultured Retinal Pigment Epithelial Cells”
Reviewer 3 Report
The manuscript "Comparison of Antioxidant Properties of Dehydrolutein with Lutein and Zeaxanthin, and their Effects on Retinal Pigment Epithelial Cells"
describes the antioxidant properties of dehydrolutein, lutein and zeaxanthin in ARPE-19 retinal cell line.
The manuscript is well written and organized, although carotenoids are very not useful in drig research. The authors should add the chemical structures of the molecules in a figure.
They also have to enrich the introduction with other articles/reviews in which natural products are used to treat retinal pathologies, such:
https://www.mdpi.com/2072-6643/5/7/2646
The discussion section could benefit also of some information about the role of carotenoids in other retinal pathologies, in order to improve the quality of the ms.
Overall, the manuscript seems to be of interest for Antioxidants and acceptable afte minor revisions.
Author Response
Response to Reviewer 3:
The manuscript "Comparison of Antioxidant Properties of Dehydrolutein with Lutein and Zeaxanthin, and their Effects on Retinal Pigment Epithelial Cells"
describes the antioxidant properties of dehydrolutein, lutein and zeaxanthin in ARPE-19 retinal cell line.
The manuscript is well written and organized, although carotenoids are very not useful in drig research. The authors should add the chemical structures of the molecules in a figure.
The AREDS and AREDS2 clinical studies were separate 5 year studies carried out by the National Eye Institute and were large randomized, multicentre, double-masked, placebo-controlled clinical trials each recruiting >4000 AMD patients. These demonstrated the benefits of an antioxidant supplement during the 5 year study periods [1,2]. More recently, lutein and zeaxanthin have been identified within the brain and associated with a cognitive benefit and are currently under further investigated as treatments to slow the cognitive decline in ageing and dementia [3-6]. As stated in the Introduction, increasing macular pigment density has been demonstrated to improve several aspects of the visual function in addition to their photoprotection function [7-18]. Carotenoids are unsurpassed singlet oxygen quenchers and accumulate in the retina in remarkably high concentrations (> 1mM) thereby having the potential to inhibit damage induced in the retina by blue light generate singlet oxygen. Understanding the high specificity and selectivity of carotenoid uptake and accumulation mechanisms is essential to developing and improving formulations that deliver these carotenoids efficiently to the retina. An additional benefit of the study of carotenoids uptake and function in ARPE-19 cells is the potential that this research can provide novel insights into the delivery across the retina-blood and brain-blood barrier that may open avenues for the delivery of therapeutics to retinal and brain tissues and the treatment of a range of pathologies . These considerations have been added to the Discussion, lines 802-823. A figure with structures of dehydrolutein, lutein and zeaxanthin has been added as Figure 1.
They also have to enrich the introduction with other articles/reviews in which natural products are used to treat retinal pathologies, such:
https://www.mdpi.com/2072-6643/5/7/2646
We have referred in the Introduction to a number of excellent reviews about macular carotenoids, some of which are considering their role in normal retinal function, and retinal pathologies such as AMD. There are a great number of natural products which have been considered for treatment of retinal pathologies and reviewing them all will lead to a very lengthy review which is beyond the scope of this presentation of new research. The excellent review suggested by the reviewer concerns vitamin A and provitamin A carotenoids which are precursors of vitamin A. The xanthophylls lutein and zeaxanthin are hydroxylated carotenoids and cannot be cleaved by the 15,15-beta-carotene dioxygenase responsible for conversion of beta-ionone ring-containing provitamin A carotenoid into vitamin A and as such albeit informative and comprehensive is only perifpheral to the research described in our paper.
The discussion section could benefit also of some information about the role of carotenoids in other retinal pathologies, in order to improve the quality of the ms.
We agree. In addition to AMD, we mentioned Stargardt’s disease and retinitis pigmentosa as diseases with potentially increased risk of photooxidative damage where hydroxyl carotenoids may provide some benefit (lines 785-786). In the revised manuscript we have expanded the Discussion and have included brief reference to a number of other diseases, in addition to AMD, associated with increased (photo)oxidative stress and/or lutein/zeaxanthin, namely Stargardt’s disease, retinitis pigmentosa, diabetic retinopathy, glaucoma, Sj¨ogren-Larsson syndrome, and macular telangiectasia, where carotenoids like zeaxanthin, lutein and dehydrolutein may be considered in development of potential therapies (lines 832-834).
Overall, the manuscript seems to be of interest for Antioxidants and acceptable afte minor revisions.
References:
- Age-Related Eye Disease Study 2 Research, G.; Chew, E.Y.; Clemons, T.E.; Sangiovanni, J.P.; Danis, R.P.; Ferris, F.L., 3rd; Elman, M.J.; Antoszyk, A.N.; Ruby, A.J.; Orth, D., et al. Secondary analyses of the effects of lutein/zeaxanthin on age-related macular degeneration progression: AREDS2 report No. 3. JAMA Ophthalmol. 2014, 132, 142-149, doi:10.1001/jamaophthalmol.2013.7376.
- Age-Related Eye Disease Study 2 Research, G.; Chew, E.Y.; SanGiovanni, J.P.; Ferris, F.L.; Wong, W.T.; Agron, E.; Clemons, T.E.; Sperduto, R.; Danis, R.; Chandra, S.R., et al. Lutein/zeaxanthin for the treatment of age-related cataract: AREDS2 randomized trial report no. 4. JAMA Ophthalmol. 2013, 131, 843-850, doi:10.1001/jamaophthalmol.2013.4412.
- Chew, E.Y.; Clemons, T.E.; Agron, E.; Launer, L.J.; Grodstein, F.; Bernstein, P.S.; Age-Related Eye Disease Study 2 Research, G. Effect of Omega-3 Fatty Acids, Lutein/Zeaxanthin, or Other Nutrient Supplementation on Cognitive Function: The AREDS2 Randomized Clinical Trial. JAMA 2015, 314, 791-801, doi:10.1001/jama.2015.9677.
- Lindbergh, C.A.; Mewborn, C.M.; Hammond, B.R.; Renzi-Hammond, L.M.; Curran-Celentano, J.M.; Miller, L.S. Relationship of Lutein and Zeaxanthin Levels to Neurocognitive Functioning: An fMRI Study of Older Adults. J. Int. Neuropsychol. Soc. 2017, 23, 11-22, doi:10.1017/S1355617716000850.
- Feeney, J.; O'Leary, N.; Moran, R.; O'Halloran, A.M.; Nolan, J.M.; Beatty, S.; Young, I.S.; Kenny, R.A. Plasma Lutein and Zeaxanthin Are Associated With Better Cognitive Function Across Multiple Domains in a Large Population-Based Sample of Older Adults: Findings from The Irish Longitudinal Study on Aging. J. Gerontol. Ser. A-Biol. Sci. Med. Sci. 2017, 72, 1431-1436, doi:10.1093/gerona/glw330.
- Nolan, J.M.; Mulcahy, R.; Power, R.; Moran, R.; Howard, A.N. Nutritional Intervention to Prevent Alzheimer's Disease: Potential Benefits of Xanthophyll Carotenoids and Omega-3 Fatty Acids Combined. J. Alzheimers Dis. 2018, 64, 367-378, doi:10.3233/JAD-180160.
- Bone, R.A.; Landrum, J.T.; Hime, G.W.; Cains, A.; Zamor, J. Stereochemistry of the human macular carotenoids. Invest. Ophthalmol. Vis. Sci. 1993, 34, 2033-2040.
- Li, B.X.; George, E.W.; Rognon, G.T.; Gorusupudi, A.; Ranganathan, A.; Chang, F.Y.; Shi, L.J.; Frederick, J.M.; Bernstein, P.S. Imaging lutein and zeaxanthin in the human retina with confocal resonance Raman microscopy. Proc. Natl. Acad. Sci. U. S. A. 2020, 117, 12352-12358, doi:10.1073/pnas.1922793117.
- Widomska, J.; SanGiovanni, J.P.; Subczynski, W.K. Why is zeaxanthin the most concentrated xanthophyll in the central fovea? Nutrients 2020, 12, doi:10.3390/nu12051333.
- Davey, P.G.; Henderson, T.; Lem, D.W.; Weis, R.; Amonoo-Monney, S.; Evans, D.W. Visual function and macular carotenoid changes in eyes with retinal drusen-An open label randomized controlled trial to compare a micronized lipid-based carotenoid liquid supplementation and AREDS-2 formula. Nutrients 2020, 12, doi:10.3390/nu12113271.
- Davey, P.G.; Lievens, C.; Ammono-Monney, S. Differences in macular pigment optical density across four ethnicities: a comparative study. Ther. Adv. Ophthalmol. 2020, 12, doi:10.1177/2515841420924167.
- Stringham, J.M.; O'Brien, K.J.; Stringham, N.T. Contrast sensitivity and lateral inhibition are enhanced with macular carotenoid supplementation. Invest. Ophthalmol. Vis. Sci. 2017, 58, 2291-2295, doi:10.1167/iovs.16-21087.
- Akuffo, K.O.; Beatty, S.; Peto, T.; Stack, J.; Stringham, J.; Kelly, D.; Leung, I.; Corcoran, L.; Nolan, J.M. The impact of supplemental antioxidants on visual function in nonadvanced age-related macular degeneration: A head-to-head randomized clinical trial. Invest. Ophthalmol. Vis. Sci. 2017, 58, 5347-5360, doi:10.1167/iovs.16-21192.
- Bernstein, P.S.; Li, B.X.; Vachali, P.P.; Gorusupudi, A.; Shyam, R.; Henriksen, B.S.; Nolan, J.M. Lutein, zeaxanthin, and meso-zeaxanthin: The basic and clinical science underlying carotenoid-based nutritional interventions against ocular disease. Prog. Retin. Eye Res. 2016, 50, 34-66, doi:10.1016/j.preteyeres.2015.10.003.
- Nolan, J.M.; Power, R.; Stringham, J.; Dennison, J.; Stack, J.; Kelly, D.; Moran, R.; Akuffo, K.O.; Corcoran, L.; Beatty, S. Enrichment of macular pigment enhances contrast sensitivity in subjects free of retinal disease: Central Retinal Enrichment Supplementation Trials - Report 1. Invest. Ophthalmol. Vis. Sci. 2016, 57, 3429-3439, doi:10.1167/iovs.16-19520.
- Barker, F.M., 2nd; Snodderly, D.M.; Johnson, E.J.; Schalch, W.; Koepcke, W.; Gerss, J.; Neuringer, M. Nutritional manipulation of primate retinas, V: effects of lutein, zeaxanthin, and n-3 fatty acids on retinal sensitivity to blue-light-induced damage. Invest. Ophthalmol. Vis. Sci. 2011, 52, 3934-3942, doi:10.1167/iovs.10-5898.
- Stringham, J.M.; Hammond, B.R. Macular pigment and visual performance under glare conditions. Optom Vis Sci 2008, 85, 82-88, doi:10.1097/OPX.0b013e318162266e.
- Stringham, J.M.; Hammond, B.R., Jr. Dietary lutein and zeaxanthin: possible effects on visual function. Nutr Rev 2005, 63, 59-64, doi:10.1111/j.1753-4887.2005.tb00122.x.
Round 2
Reviewer 2 Report
All comments are adressed.